# Discrete Diffusion Samplers and Bridges:
# Off-Policy Algorithms and Applications in Latent Spaces

**Arran Carter** [1] [*]   **Sanghyeok Choi** [1] [*]   **Kirill Tamogashev** [1] [*]   **Víctor Elvira** [1]   **Esmeralda S. Whitammer** [1] [2]

 mmacosha/offpolicy-discrete-diffusion-samplers-and-bridges

## Abstract

Sampling from a distribution $p(x) \propto e^{-\mathcal{E}(x)}$ known up to a normalising constant is an important and challenging problem in statistics. Recent years have seen the rise of a new family of amortised sampling algorithms, commonly referred to as diffusion samplers, that enable fast and efficient sampling from an unnormalised density. Such algorithms have been widely studied for continuous-space sampling tasks; however, their application to problems in discrete space remains largely unexplored. Although some progress has been made in this area, discrete diffusion samplers do not take full advantage of ideas commonly used for continuous-space sampling. In this paper, we propose to bridge this gap by introducing off-policy training techniques for discrete diffusion samplers. We show that these techniques improve the performance of discrete samplers on both established and new synthetic benchmarks. Next, we generalise discrete diffusion samplers to the task of bridging between two arbitrary distributions, introducing data-to-energy Schrödinger bridge training for the discrete domain for the first time. Lastly, we showcase the application of the proposed diffusion samplers to data-free posterior sampling in the discrete latent spaces of image generative models.

## 1. Introduction

In this paper, we consider the problem of sampling from a distribution over a discrete space $\mathcal{S} = \{1, \ldots, C\}^d$ with probability mass function $p_{\text{target}}(x) = \frac{1}{Z} e^{-\mathcal{E}(x)}$. The energy

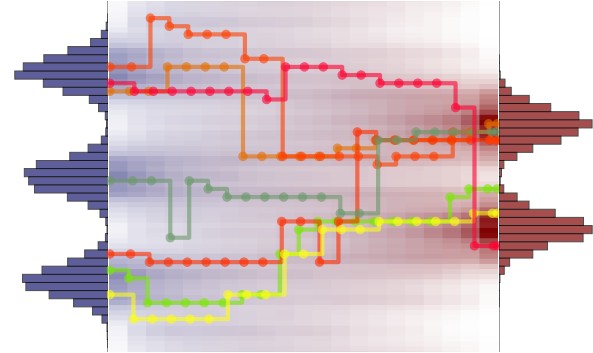

*Figure 1.* Trajectories sampled from a learnt approximation to the discrete Schrödinger bridge between a mixture of three Gaussians (left, given by samples), and a mixture of two Gaussians (right, given by an unnormalised density), both quantised and represented as 6-bit binary Gray codes. Background shading represents the marginal densities at each step.

function $\mathcal{E} : \mathcal{S} \rightarrow \mathbb{R}$ can be queried pointwise, but the normalising constant $Z$ is not known and samples from $p_{\text{target}}$ cannot tractably be drawn. This is the typical setting in variational (Bayesian) inference, and it also appears in statistical mechanics, where energy-based models over discrete spaces are considered (Ising, 1925; Potts, 1952).

Traditionally, sampling from a distribution defined by an unnormalised density is done using Monte Carlo (MC) methods, which exist both for continuous and discrete spaces. Algorithms for densities defined on a continuous domain include Metropolis-Hastings Markov Chain MC (MCMC; Metropolis et al., 1953), annealed importance sampling (Neal, 2001), sequential MC (Del Moral et al., 2006), adaptive importance sampling (Bugallo et al., 2017), Hamiltonian MC (Neal et al., 2011), as well as their variations. Their discrete analogues include MCMC on discrete state spaces, such as Metropolis-Hastings and Gibbs sampling, with foundational methodological work by Hastings (1970), Besag (1974), and Geman & Geman (1984), as well as theoretical developments for sampling and approximate counting in combinatorial spaces by Jerrum & Sinclair (1996). Gradient-based MCMC methods in the discrete domain have also been explored (Grathwohl et al., 2021; Zhang et al., 2022b).

---

[*]Equal contribution [1]University of Edinburgh [2]CIFAR Fellow. Correspondence to: Arran Carter <arran.carter@ed.ac.uk>.

*Proceedings of the 43rd International Conference on Machine Learning*, Seoul, South Korea. PMLR 306, 2026. Copyright 2026 by the author(s).

The popularity of diffusion generative models as approximators for empirical data distributions (Sohl-Dickstein et al., 2015; Ho et al., 2020; Song et al., 2021) motivated the adaptation of similar techniques to sampling problems, giving rise to the family of sampling algorithms called *diffusion samplers*. Diffusion samplers (*e.g.*, Zhang & Chen, 2022; Vargas et al., 2023; Richter et al., 2024; Chen et al., 2025a; Albergo & Vanden-Eijnden, 2025) use a diffusion process to model the transition dynamics from a tractable distribution to a target distribution from which samples are not available. Unlike diffusion generative models, which are trained to maximise a variational bound on the likelihood of true samples, diffusion samplers typically optimise a stochastic optimal control (Zhang & Chen, 2022; Berner et al., 2024) or path-space matching (Richter et al., 2020; Lahlou et al., 2023) objective; see Berner et al. (2026) for unifying theory. These samplers can be further improved by the introduction of off-policy reinforcement learning (RL) techniques to facilitate mode coverage and improve sampling quality (Sendera et al., 2024; Choi et al., 2026).

Recently, diffusion samplers have also been adapted for sampling from discrete unnormalised densities (Holderrieth et al., 2025; Zhu et al., 2025; Sanokowski et al., 2025a), building upon the discrete diffusion modelling framework established by Hoogeboom et al. (2021); Austin et al. (2021); Campbell et al. (2022). However, these algorithms do not take full advantage of off-policy RL. At the same time, off-policy algorithms equivalent to discrete diffusion samplers were considered as early as Zhang et al. (2022a), but the connection has not been recognised until now. Encouraged by these successes, we make our first contribution:

(1) We introduce off-policy RL methods into the training of discrete diffusion samplers (§2.2) and show their benefits on various established and new tasks (§5.1 and 5.2).

We also consider the problem of bridging two arbitrary distributions by finding the Schrödinger bridge (Schrödinger, 1931; 1932), a dynamic version of an optimal transport plan. This problem has been solved for continuous (De Bortoli et al., 2021; Vargas et al., 2021, *inter alia*) and discrete (Kim et al., 2025a; Ksenofontov & Korotin, 2025) cases. However, these solutions consider the setting when samples for both distributions are available – the data-to-data setting. Recently, the problem of finding the Schrödinger bridge between two distributions when one (or both) of them is only given by an unnormalised density function (data-to-energy and energy-to-energy, respectively) was considered by Tamogashev & Malkin (2026). Building upon their work:

(2) We generalise discrete diffusion samplers to data-to-energy Schrödinger bridges (§3 and §5.2; Fig. 1).

Previously, evaluation of discrete diffusion samplers has mostly been conducted on synthetic densities. We show, for the first time, that discrete diffusion samplers can enable posterior inference in the discrete latent spaces of pretrained image generative models, forming our third contribution:

(3) We apply discrete diffusion samplers to the problem of sampling posterior distributions in the latent spaces of generative models (§4 and §5.3).

## 2. Algorithms for Discrete Diffusion Samplers

We now describe the preliminaries (§2.1) and new methods (§2.2) for discrete diffusion sampling. Subsequently, we generalise the sampling problem to the problem of bridging between two arbitrary distributions (§3). **Related work** is discussed throughout the text as relevant and further in §6.

### 2.1. Setting and Preliminaries

The exposition here synthesises the views of discrete diffusion via any-order autoregressive modelling and stochastic optimal control (Austin et al., 2021; Zhu et al., 2025) in a way that makes clear their connection to hierarchical variational inference (Ranganath et al., 2016) and the analogy with the theory for continuous-space diffusion samplers (Berner et al., 2026).

We wish to sample from the discrete state space $\mathcal{S} = \{1, \ldots, C\}^d$ of sequences of length $d$ in a vocabulary of size $C$ according to the distribution with probability mass function $p_{\text{target}}(x) = \frac{1}{Z} e^{-\mathcal{E}(x)}$, where $\mathcal{E} : \mathcal{S} \to \mathbb{R}$ is an energy function. We do not assume knowledge of the normalising constant $Z = \sum_{x \in \mathcal{S}} e^{-\mathcal{E}(x)}$.

**Sampling with discrete measure transport.** Discrete diffusion models decompose sampling from $\mathcal{S}$ as a Markov chain $X_0 \to X_1 \to \cdots \to X_N$, where each $X_n$ is a random variable taking values in $\mathcal{S}$ or in some extended space $\overline{\mathcal{S}} \supseteq \mathcal{S}$ (for example, in masked diffusion, $\overline{\mathcal{S}} = (\{1, \ldots, C\} \cup \{\star\})^d$, where $\star$ is the mask token). The goal is to find transition kernels[1] $\overrightarrow{p}_\theta(X_{n+1} \mid X_n)$ such that when $X_0$ is distributed as some known distribution $p_0$, the marginal distribution over $X_N$ induced by following the kernels for $N$ steps has support only over $\mathcal{S}$ and coincides on $\mathcal{S}$ with the target $p_{\text{target}}(X_N)$.

The initial distribution $p_0$ and kernels $\overrightarrow{p}_\theta$ determine a distribution over trajectories $X_{0:N} = (X_0, \ldots, X_N)$:

$$(p_0 \otimes \overrightarrow{p}_\theta^{\otimes N})(X_{0:N}) = p_0(X_0) \prod_{n=0}^{N-1} \overrightarrow{p}_\theta(X_{n+1} \mid X_n). \quad (1)$$

Restating the above goal, we aim for the marginal distribution $(p_0 \otimes \overrightarrow{p}_\theta^{\otimes N})(X_N)$ to be equal to $p_{\text{target}}(X_N)$.

In order to set a learning target for the forward kernels $\overrightarrow{p}_\theta$, we define backward (noising) kernels $\overleftarrow{p}(X_n \mid X_{n+1})$, which

---

[1]We do not explicitly write the dependence on the time step $n$ in the kernel.

similarly give a distribution over trajectories

$$(p_{\text{target}} \otimes \overleftarrow{p}^{\otimes N})(X_{0:N}) = p_{\text{target}}(X_N) \prod_{n=0}^{N-1} \overleftarrow{p}(X_n \mid X_{n+1}), \ (2)$$

where $p_{\text{target}}$ is understood to be 0 for elements of $\overline{\mathcal{S}} \setminus \mathcal{S}$.

Learning objectives for discrete diffusion samplers aim to match the two distributions over trajectories (1) and (2) in some measure of divergence. If the two distributions coincide, their marginal distributions over $X_N$ coincide; because $(p_{\text{target}} \otimes \overleftarrow{p}^{\otimes N})(X_N) = p_{\text{target}}(X_N)$ by construction, satisfying this equality achieves the desired goal.

**Common choices of $\overleftarrow{p}$.** Two common choices for $\overleftarrow{p}$, introduced by Austin et al. (2021), are the uniform diffusion process and the masking process, where each transition $\overleftarrow{p}(X_n \mid X_{n+1})$ replaces the entries at some randomly chosen positions in $X_{n+1}$ by the mask token $\star$ (see §A).

To make it possible to match the forward and reverse distributions, the marginal $(p_{\text{target}} \otimes \overleftarrow{p}^{\otimes N})(X_0)$ should equal $p_0(X_0)$. For example, in the case of masking diffusion, $p_0$ and the final kernel $\overleftarrow{p}(X_0 \mid X_1)$ both put all mass on the state $(\star, \ldots, \star)$. Similarly, we require that the final kernel $\overrightarrow{p}_\theta(X_N \mid X_{N-1})$ places zero mass on $\overline{\mathcal{S}} \setminus \mathcal{S}$.

We remark that if $\overleftarrow{p}$ masks exactly one entry on each transition, we precisely recover the setting of Zhang et al. (2022a), whose backward policy is used as an auxiliary object in training samplers for energy-based models. Those algorithms, described there in the language of generative flow networks (Bengio et al., 2021; 2023), used trajectory balance ((3) below) and off-policy training (§2.2). It should be noted that this line of work – which draws more from the entropic RL literature (cf. Tiapkin et al. (2024); Deleu et al. (2024)) than from the domain of generative modelling – predates the recent developments in masked discrete diffusion samplers (Zhu et al., 2025; Sanokowski et al., 2025a), yet the connection has not been recognised.

**Objectives.** Various loss functions exist to enforce the matching of distributions induced by the forward and backward processes. Those we study are of the second-moment form, depending on a distribution over trajectories $\mathbb{P}$:

$$\mathcal{L}^{\mathbb{P}} = \mathbb{E}_{X_{0:N} \sim \mathbb{P}} \left[ \left( \log \frac{(p_0 \otimes \overrightarrow{p}_\theta^{\otimes N})(X_{0:N})}{(p_{\text{target}} \otimes \overleftarrow{p}^{\otimes N})(X_{0:N})} - c \right)^2 \right], \quad (3)$$

where the numerator and denominator are given by (1) and (2), respectively, and $c$ is a scalar that we choose.

Note that while the mass function $p_{\text{target}}(X_N) \propto e^{-\mathcal{E}(X_N)}$ is only known up to a normalising constant, this constant can be additively absorbed into $c$. Taking $c$ to be a learnt parameter recovers the *trajectory balance* (TB) objective, which was originally introduced for discrete-space sampling in

Malkin et al. (2022) and used for a variety of discrete-space sampling problems (Jain et al., 2022; Deleu et al., 2022; van Krieken et al., 2023; Kim et al., 2024, *inter alia*), while taking $c$ to be the value minimising the empirical mean over a batch of samples from $\mathbb{P}$ recovers the *log-variance* (LV) objective, introduced in Richter et al. (2020) and subsequently studied for both continuous-space (Richter et al., 2024; Sendera et al., 2024; Gritsaev et al., 2025; Blessing et al., 2025) and discrete-space diffusion (Holderrieth et al., 2025; Zhu et al., 2025; Guo et al., 2026).

Notably, if $\mathbb{P} = p_0 \otimes \overrightarrow{p}_\theta^{\otimes N}$ (*on-policy objective*), the gradient of $\mathcal{L}^{\mathbb{P}}$ coincides up to scalar with the gradient of the (reverse) KL divergence for any choice of $c$ (Richter et al., 2020; Malkin et al., 2023; Zimmermann et al., 2023):

$$\nabla_\theta \mathcal{L}^{\mathbb{P}}|_{\mathbb{P}=p_0 \otimes \overrightarrow{p}_\theta^{\otimes N}} = 2\nabla_\theta \text{KL}(p_0 \otimes \overrightarrow{p}_\theta^{\otimes N} \parallel p_{\text{target}} \otimes \overleftarrow{p}^{\otimes N}).$$

However, the minimiser of (3) for *any* full-support $\mathbb{P}$ is unique and satisfies $p_0 \otimes \overrightarrow{p}_\theta^{\otimes N} = p_{\text{target}} \otimes \overleftarrow{p}^{\otimes N}$, which allows the use of other distributions $\mathbb{P}$, possibly varying over the course of training. The choice of $\mathbb{P}$ is the subject of the next subsection.

**Variable time discretisation.** Continuous-time diffusion processes can be sampled with varying time discretisation, effectively modifying the step size in the Euler-Maruyama integration of a stochastic differential equation (Song et al., 2021). Similarly, the continuous-time view of discrete diffusion (Campbell et al., 2022) allows sampling from a denoising process with a larger number of steps than used during training by changing the transition kernel appropriately while keeping the model fixed. Extending the results of Berner et al. (2026) to the discrete-space case, we find that unmasking diffusion samplers trained with few unmasking steps perform well when sampled with one unmasked position per transition; see §A.1 for details.

## 2.2. Off-Policy Training Strategies

The choice of $\mathbb{P}$ gives rise to two different ways in which to train models: the choice $\mathbb{P} = p_0 \otimes \overrightarrow{p}_\theta^{\otimes N}$ is referred to as "on-policy" training. Off-policy training – that is, a choice of $\mathbb{P}$ that differs from $p_0 \otimes \overrightarrow{p}_\theta^{\otimes N}$ – allows for efficient exploration. Many varieties of off-policy trajectory distributions $\mathbb{P}$ have been considered in the literature on RL for sampling or on generative flow networks, ranging from application-specific heuristic search (Phillips & Cipcigan, 2025; Kim et al., 2024; 2025b) to learnt exploration policies and approximate Bayesian ensembles (Kim et al., 2025c; Rector-Brooks et al., 2023; Muhammad & Lahlou, 2025).

We describe three off-policy strategies:

**Replay buffer.** A replay buffer is a common reinforcement learning technique (Lin, 1992; Mnih, 2013) in which samples drawn from the policy are stored and reused at a

later training step. In our setting, following Sendera et al. (2024), we maintain a buffer of sampled states $X_N$ and reuse them by sampling from $\overleftarrow{p}^{\otimes N}$ to produce trajectories to train with. This procedure mixes samples from the current model and from its earlier iterations, which can be particularly useful if the predictive distribution is rapidly changing during the course of training.

**Importance-weighted buffer.** Instead of simply sampling from the replay buffer with uniform probability for all elements in the buffer, we can assign different sampling probabilities to each buffer element, known as prioritisation (Schaul et al., 2016). Such a scheme was considered for amortised samplers by Tiapkin et al. (2024); Shen et al. (2023). Choi et al. (2026) proposed to weight samples by the importance weights $w$ of the trajectories that produced the buffer samples, computed *at the time the samples are added to the buffer* as:

$$w = \frac{e^{-\mathcal{E}(X_N)} \prod_{n=0}^{N-1} \overleftarrow{p}(X_n \mid X_{n+1})}{p_0(X_0) \prod_{n=0}^{N-1} \overrightarrow{p}_\theta(X_{n+1} \mid X_n)}, \tag{4}$$

where $X_{0:N}$ is a trajectory sampled from $p_0 \otimes \overrightarrow{p}_\theta^{\otimes N}$. By using importance-weighted prioritisation, we focus the training onto samples obtained along trajectories that had high target probability mass but which the sampler had a lower probability of producing.

**MCMC exploration from the buffer.** In addition to relying on the evolving sampler to explore the space, one can use local search techniques, such as Markov chain Monte Carlo (MCMC), to explore $p_{\text{target}}$ and guide the training of the sampler to regions of high density. We adapt the methods for continuous diffusion samplers (Sendera et al., 2024) to discrete spaces, although Monte Carlo methods have been combined with amortised sequential samplers in various other ways (Huang et al., 2024; Akhound-Sadegh et al., 2024; Vargas et al., 2024; Choi et al., 2026; Wu et al., 2025).

Specifically, this is done by refining samples from the replay buffer with several steps of an MCMC kernel whose stationary distribution is $p_{\text{target}}$ before they are eventually used for training. Because MCMC requires only evaluations of the energy function, not the model, it is computationally inexpensive compared to sampler rollouts. The MCMC kernel should be chosen appropriately for the target energy (we explore different options in §5.1 and D.1.3). Algorithm 1 describes the full training procedure.

## 3. Discrete Data-to-Energy Schrödinger Bridges

The bridge problem is a generalisation of the sampling problem described in the previous sections. Rather than taking $p_0$ to be a simple, known distribution and choosing $\overleftarrow{p}$ such

that $(p_{\text{target}} \otimes \overleftarrow{p}^{\otimes N})(X_0) = p_0(X_0)$ by construction, we take $p_0$ to be arbitrary and learn $\overleftarrow{p}_\varphi$ as a parametric kernel. The bridge problem seeks a pair of time-dependent kernels $\overrightarrow{p}_\theta$ and $\overleftarrow{p}_\varphi$ that represent the same joint distribution over $X_{0:N}$ transporting $p_0$ to $p_N = p_{\text{target}}$ and vice versa, *i.e.*, satisfying $p_0 \otimes \overrightarrow{p}_\theta^{\otimes N} = p_{\text{target}} \otimes \overleftarrow{p}_\varphi^{\otimes N}$. (Here, we assume that $\overline{\mathcal{S}} = \mathcal{S}$.)

Among all pairs of kernels that solve the bridge problem, the Schrödinger bridge (SB; Schrödinger, 1931; 1932) is the pair that minimises the KL divergence to some reference distribution $\mathbb{Q}$, assumed to be given by a kernel $\overrightarrow{p}_{\text{ref}}$:[2]

$$(\overrightarrow{p}^*, \overleftarrow{p}^*) = \underset{\substack{(\overrightarrow{p}, \overleftarrow{p}): \\ \mathbb{P} := p_0 \otimes \overrightarrow{p}^{\otimes N} = p_{\text{target}} \otimes \overleftarrow{p}^{\otimes N}}}{\arg\min} \; \mathrm{KL}(\mathbb{P} \parallel \mathbb{Q}). \tag{5}$$

The SB problem has been studied in the case where samples from both $p_0$ and $p_N$ are available, both in the continuous-space (De Bortoli et al., 2021; Vargas et al., 2021; Chen et al., 2021; 2022; Shi et al., 2023; Tong et al., 2024) and discrete-space (Kim et al., 2025a; Ksenofontov & Korotin, 2025) cases. Recently, Tamogashev & Malkin (2026) developed a SB algorithm for the setting where there are no samples available from $p_N$ and only an energy function is given. This algorithm is an adaptation of the iterative proportional fitting procedure (IPF; Fortet, 1940; Kullback, 1968; Chen et al., 2021), studied in the data-to-data case by De Bortoli et al. (2021); Vargas et al. (2021), that uses off-policy objectives similar to those described in §2.1. This adaptation was studied for the continuous state space setting; here we investigate its extension into discrete state spaces.

IPF alternately fits joint distributions $\overrightarrow{\mathbb{P}}$ and $\overleftarrow{\mathbb{P}}$. Initialising $\overrightarrow{\mathbb{P}}^0 = \mathbb{Q}$, we iteratively solve:

$$\overleftarrow{\mathbb{P}}^{k+1} = \underset{\overleftarrow{\mathbb{P}}: \overleftarrow{\mathbb{P}}(X_0) = p_0(X_0)}{\arg\min} \; \mathrm{KL}(\overleftarrow{\mathbb{P}} \mid \overrightarrow{\mathbb{P}}^k) \tag{6a}$$

$$\overrightarrow{\mathbb{P}}^{k+1} = \underset{\overrightarrow{\mathbb{P}}: \overrightarrow{\mathbb{P}}(X_N) = p_N(X_N)}{\arg\min} \; \mathrm{KL}(\overrightarrow{\mathbb{P}} \mid \overleftarrow{\mathbb{P}}^{k+1}) \tag{6b}$$

for $k = 0, 1, 2, \dots$. As $k \to \infty$, the two processes can be shown to converge (in the KL sense) to the solution of (5). The solution at each iteration is Markov, and we assume $\overrightarrow{\mathbb{P}}^k, \overleftarrow{\mathbb{P}}^k$ to be represented by kernels $\overrightarrow{p}^k, \overleftarrow{p}^k$, respectively.

Choosing a parametric representation of the kernels, denoted $\overrightarrow{p}_\theta^k, \overleftarrow{p}_\varphi^k$, the optimisation problems in (6) can be solved approximately. When samples from $p_0$ are available, (6a) can be solved by maximum-likelihood training. However, when samples from $p_N$ are not available, (6b) features the problem of exploration of $p_{\text{target}}$ and requires a variant of the log-variance loss (see Tamogashev & Malkin,

---

[2]Although the SB problem is typically posed as a minimisation over all processes (joint distributions), the solution can be shown to be Markov and thus factorisable into kernels in either direction.

2026, equation (8), for derivation). This objective can be directly translated to our discrete-space setting, and the full algorithm is detailed in Algorithm 2.

# 4. Outsourced Samplers and Bridges in Discrete Latent Spaces

Constraining, or conditioning, a pretrained model is a fundamental problem of generative modelling. Conditional sampling can be formulated as posterior inference: given a pretrained model $p(x)$ and a condition $p(y \mid x)$ – such as the likelihood under a classifier of $x$ belonging to class $y$ – the goal is to sample from the posterior $p(x \mid y) \propto p(x)p(y \mid x)$ without observing any unbiased conditional samples.

If the prior $p(x)$ can be expressed as the pushforward by a function $f$ of a random noise variable $z \sim p_{\text{latent}}(z)$[3], then the posterior sampling problem is equivalent to sampling $z$ from $p(z \mid y) \propto p_{\text{latent}}(z)p(y \mid f(z))$. The samples $f(z)$ for $z \sim p(z \mid y)$ then follow the posterior $p(x \mid y)$. This observation was used by Venkatraman et al. (2025), who proposed to train diffusion samplers of posteriors $p(z \mid y)$ in the (continuous) latent spaces of various generative models, such as VAEs, GANs, and continuous normalising flows. Such an approach to posterior inference does not require evaluating the density of $p(x)$ or differentiating through $f$. (Alternative methods for drawing samples from $p(z \mid y)$, also using this *noise outsourcing* principle, were proposed in Tang et al. (2025); Wang et al. (2026); Kalaivanan et al. (2025); Om et al. (2025).)

Generalising the setup introduced in Venkatraman et al. (2025) to discrete latent spaces, we can train diffusion samplers to model the Bayesian posterior distribution $p(z \mid y)$ in the latent space of a pretrained VQ-VAE model (Van Den Oord et al., 2017) with autoregressive prior $p_{\text{latent}}(z)$, which is a model over token sequences of fixed length. One could similarly model Schrödinger bridges between posterior distributions in latent space defined in this way.

# 5. Experiments

We empirically evaluate the off-policy sampler training strategies (§2.2), as well as their Schrödinger bridge generalisation (§3), on synthetic densities in §5.1 and 5.2 and on outsourced posterior sampling (§4) in §5.3.

**Modelling setting: sampling.** We train discrete diffusion samplers $\overrightarrow{p}_\theta$ under a masking noising kernel $\overleftarrow{p}$ (details in §A.1). This setup coincides with that formulated in the stochastic optimal control setting by MDNS (Zhu et al., 2025), which primarily used the weighted denoising cross-entropy (WDCE) loss for high-dimensional problems and only considered a masking kernel $\overleftarrow{p}$ that masks one en-

---

[3]That is, if $z \sim p_{\text{latent}}(z)$ and $x = f(z)$, then $x \sim p(x)$.

try per step. In contrast, we allow for multiple maskings per step in $\overleftarrow{p}$ (and correspondingly multiple unmaskings in $\overrightarrow{p}_\theta$); see §D.1.4 for a study of the effect. As known from the literature on discrete diffusion models trained on target data (Hayakawa et al., 2025; Chen et al., 2025b), this introduces correlation error, but increases computational speed and eases memory requirements for many choices of loss function. We use the methods of MDNS as our main baseline to which we compare second-moment divergences optimised by both on- and off-policy methods on synthetic targets.

We compare samplers trained with TB and LV objectives (§2.1), either with on-policy training or with some combination of the importance-weighted buffer and MCMC exploration strategies from §2.2. We include results for a (learning-free) Metropolis-Hastings algorithm for comparison to the MDNS baseline and to the samplers we study.

**Modelling setting: bridges.** Masked diffusion is not applicable to the Schrödinger bridge problem. Instead, we take a uniform diffusion process (Austin et al., 2021), detailed in §A.2, as the reference kernel $\overrightarrow{p}_{\text{ref}}$.

## 5.1. Ising and Potts Models

The **Potts model** (Potts, 1952) is an energy-based model from statistical mechanics, defined over a space of *spin states* – discrete variables taking values in $\{1, \ldots, q\}$. We consider spins spatially arranged in an $L \times L$ toroidal lattice, so the state space is $\mathcal{S} = \{1, \ldots, q\}^{L \times L}$.

The energy function depends on a spin interaction coefficient, which we take to be 1 by default. The energy and probability mass functions are then

$$H(x) = -J \sum_{i \sim j} \mathbf{1}_{x_i = x_j}, \quad p_{\text{target}}(x) \propto e^{-\beta H(x)}, \quad (7)$$

where $\beta \in \mathbb{R}^+$ is the inverse temperature and $i \sim j$ denotes that coordinates $i$ and $j$ are adjacent. When $J$ is positive, the energy model favours configurations in which adjacent spins coincide and has $q$ modes, each corresponding to a state with all spins equal. The standard **Ising model** (Ising, 1925) is a special case of the Potts model with $q = 2$, $J = \frac{1}{2}$.

We train discrete diffusion samplers on the target energies of Potts and Ising models with $L = 16$ and vary the inverse temperature $\beta$ to control the difficulty of the problem – a larger $\beta$ (lower temperature) can lead to more difficult and less stable learning, often collapsing to a single mode. Following Zhu et al. (2025), we report the empirical magnetisation and correlation errors, as well as the Sinkhorn distance between approximate and true samples, the ELBO, and the EUBO (Blessing et al., 2024); definitions are given in §C.1.3. The correlation error, Sinkhorn, and EUBO metrics require access to true samples, which were obtained from a long-run MCMC chain.

| Inverse temp. → | Ising, $\beta = 0.4407$ | | | | | Ising, $\beta = 0.6$ | | | | | Ising, $\beta = 1.2$ | | | | |
|---|---|---|---|---|---|---|---|---|---|---|---|---|---|---|---|
| Method ↓ Metric → | ELBO ↑ | EUBO ↓ | Sink. ↓ | Mag. ↓ | Corr. ↓ | ELBO ↑ | EUBO ↓ | Sink. ↓ | Mag. ↓ | Corr. ↓ | ELBO ↑ | EUBO ↓ | Sink. ↓ | Mag. ↓ | Corr. ↓ |
| MH MCMC | - | - | 48.47±0.29 | 0.04±0.01 | 0.14±0.01 | - | - | 8.64±0.21 | 0.09±0.06 | 0.22±0.02 | - | - | 1.05±0.21 | 0.06±0.04 | 0.23±0.02 |
| MDNS (WDCE) | 237.99±0.49 | 245.66±8.30 | 61.60±18.23 | 0.32±0.35 | 0.23±0.26 | 310.18±0.33 | 341.82±38.37 | 48.71±55.25 | 0.41±0.46 | 0.38±0.46 | 614.42±0.00 | 1192.26±94.09 | 126.27±0.90 | 1.00±0.00 | 1.00±0.00 |
| LV (on-policy) | 237.89±0.16 | 242.32±4.20 | 53.30±11.11 | 0.67±0.10 | 0.43±0.10 | 309.77±0.04 | 422.53±94.60 | 116.96±2.15 | 0.97±0.00 | 0.95±0.00 | 614.42±0.00 | 1726.06±144.71 | 126.89±0.96 | 1.00±0.00 | 1.00±0.00 |
| LV + Buffer | 238.21±0.24 | 239.00±0.18 | 47.80±0.62 | **0.03±0.02** | 0.07±0.02 | 308.55±1.36 | 311.05±0.36 | 4.33±0.53 | 0.40±0.30 | 0.27±0.25 | **615.07±0.05** | **615.12±0.01** | **0.02±0.00** | **0.01±0.01** | **0.00±0.00** |
| LV + Buffer + MCMC | 238.28±0.04 | 238.97±0.03 | 47.34±0.90 | 0.12±0.07 | 0.06±0.01 | 310.18±0.24 | 310.65±0.08 | 3.85±0.20 | 0.14±0.13 | 0.04±0.05 | 614.82±0.55 | 615.16±0.09 | 0.04±0.04 | 0.05±0.08 | 0.01±0.02 |
| TB (on-policy) | 237.64±0.23 | 249.26±9.65 | 68.79±16.84 | 0.78±0.02 | 0.55±0.02 | 309.79±0.00 | 377.92±14.19 | 117.22±0.68 | 0.97±0.00 | 0.95±0.00 | 614.42±0.00 | 1836.55±12.21 | 127.82±0.47 | 1.00±0.00 | 1.00±0.00 |
| TB + Buffer | _238.43±0.02_ | _238.85±0.02_ | 46.44±0.67 | 0.08±0.05 | _0.03±0.01_ | _310.42±0.03_ | _310.56±0.01_ | _3.59±0.04_ | _0.04±0.02_ | **0.00±0.00** | 614.75±0.30 | 922.79±381.43 | 50.94±62.35 | 0.42±0.47 | 0.40±0.49 |
| TB + Buffer + MCMC | **238.48±0.01** | **238.80±0.01** | **46.22±0.36** | _0.04±0.02_ | **0.02±0.01** | **310.43±0.01** | **310.55±0.01** | **3.47±0.12** | **0.02±0.01** | **0.00±0.00** | _615.03±0.06_ | _615.14±0.02_ | **0.02±0.00** | _0.03±0.01_ | **0.00±0.00** |

*Table 1.* Experimental results on $16 \times 16$ Ising (above) and Potts (right) models (mean±std over 5 runs), showing that off-policy methods deliver consistent improvements.

| Inverse temp. → | Potts, $q = 3$, $\beta = 1.005$ | | | | | Potts, $q = 3$, $\beta = 1.2$ | | | | |
|---|---|---|---|---|---|---|---|---|---|---|
| Method ↓ Metric → | ELBO ↑ | EUBO ↓ | Sink. ↓ | Mag. ↓ | Corr. ↓ | ELBO ↑ | EUBO ↓ | Sink. ↓ | Mag. ↓ | Corr. ↓ |
| MH MCMC | - | - | 90.36±1.10 | **0.03±0.01** | 0.18±0.01 | - | - | 47.27±1.29 | **0.03±0.01** | 0.28±0.00 |
| MDNS (WDCE) | 530.03±0.60 | 534.32±5.39 | 84.78±10.15 | 0.08±0.06 | **0.01±0.00** | 620.23±0.52 | 680.52±53.78 | 99.95±70.52 | 0.58±0.46 | 0.01±0.00 |
| LV (on-policy) | 529.29±0.16 | 556.12±6.21 | 136.86±7.12 | 0.81±0.01 | 0.07±0.01 | 619.82±0.10 | 748.31±70.53 | 156.71±2.69 | 0.95±0.00 | **0.00±0.00** |
| LV + Buffer | 529.26±0.70 | 532.20±0.41 | 83.32±1.92 | 0.04±0.03 | 0.08±0.02 | 619.47±1.00 | 628.47±13.62 | 32.88±29.09 | 0.21±0.17 | 0.02±0.01 |
| LV + Buffer + MCMC | 528.84±1.07 | 532.45±0.64 | 84.58±1.02 | 0.04±0.03 | 0.09±0.02 | 619.38±1.17 | _621.60±0.15_ | 19.56±5.21 | 0.13±0.10 | 0.02±0.02 |
| TB (on-policy) | 529.06±0.18 | 573.71±14.83 | 143.51±4.51 | 0.81±0.02 | 0.07±0.02 | 619.34±0.41 | 1117.43±235.46 | 161.78±3.45 | 0.96±0.01 | 0.01±0.01 |
| TB + Buffer | _530.51±0.09_ | _531.43±0.08_ | 78.89±0.98 | **0.03±0.02** | 0.02±0.01 | _619.84±0.03_ | 705.56±16.57 | 156.12±2.14 | 0.95±0.00 | **0.00±0.00** |
| TB + Buffer + MCMC | **530.65±0.04** | **531.31±0.03** | **78.50±0.83** | **0.03±0.02** | 0.01±0.01 | **620.73±0.15** | **621.30±0.08** | **12.37±0.82** | **0.03±0.02** | **0.00±0.00** |

*Figure 2.* Samples generated by each method for the $16 \times 16$ Potts model with $\beta = 1.005$ (top) and $\beta = 1.2$ (bottom), $q = 3$.

The off-policy methods with MCMC use the Swendsen-Wang algorithm (Swendsen & Wang, 1986; 1987) as a proposal tailored to the structure of this energy model. We note, however, that the MCMC used for exploration during training is not run for sufficiently long for the chains to converge. For a comparison of off-policy methods trained instead with a $H$-Hamming-ball proposal MH, see §D.1.3. For all other experiment details, see §C.1.

**Results.** The results on Ising models in Table 1 (top) show that the off-policy methods consistently outperform or match on-policy methods and that TB performs slightly better than LV on most temperatures. The greatest difference between on-policy and off-policy methods occurs at the highest inverse temperatures, as can be visually seen in Figs. 7 and 8 in §D.2.1. In the lowest-temperature setting ($\beta = 1.2$), we see that many of the methods experience severe mode collapse, however, both MCMC-assisted methods successfully manage to model the distribution well.

Table 1 (right) shows results on the Potts model with $q = 3$. The TB off-policy methods outperform both on-policy methods, as well as LV and MDNS. At the lowest temperature

($\beta = 1.2$), many of the methods collapse to a single mode, while both MCMC-assisted methods do not (Fig. 2, second row). Interestingly, for both Ising and Potts models, TB without MCMC is sometimes unstable at low temperatures.

## 5.2. Discretised Synthetic Densities

As challenging densities with entangled variables, we consider distributions over $\mathbb{R}^D$, which are converted to distributions over *binary* codes of length $d = 8D$ by subsampling each dimension at $2^8$ equally spaced values and representing each resulting coordinate by a Gray code (Gray (1953), §C.2.1). Such a setting was previously considered for discrete probabilistic modelling in Dai et al. (2020); Zhang et al. (2022a), but not for the problem of sampling a given energy-based model.

We make use of two common synthetic densities for continuous sampling problems, ManyWell ($D = 4$ or 10, a sum of bivariate fourth-degree polynomial energies with $2^D$ modes; Noé et al. (2019)) and 40GMM ($D = 2$ or 4, a mixture of 40 Gaussians; Midgley et al. (2023)). These continuous-space densities are highly multimodal and

*Table 2.* Experimental results on discretised synthetic densities (mean±std over 5 runs).

| Target → | 40GMM ($d = 2 \times 8 = 16$) | | | | 40GMM ($d = 4 \times 8 = 32$) | | | | ManyWell ($d = 4 \times 8 = 32$) | | | | ManyWell ($d = 10 \times 8 = 80$) | | | |
|---|---|---|---|---|---|---|---|---|---|---|---|---|---|---|---|---|
| Method ↓ Metric → | ELBO ↑ | EUBO ↓ | MMD ↓ | Sink. ↓ | ELBO ↑ | EUBO ↓ | MMD ↓ | Sink. ↓ | ELBO ↑ | EUBO ↓ | MMD ↓ | Sink. ↓ | ELBO ↑ | EUBO ↓ | MMD ↓ | Sink. ↓ |
| MH MCMC | - | - | **0.01**±0.00 | **0.04**±0.00 | - | - | 0.05±0.00 | 4.86±8.49 | - | - | **0.01**±0.00 | **0.04**±0.00 | - | - | **0.01**±0.00 | 1.06±0.01 |
| MDNS (WDCE) | -17.30±1.76 | 1.54±0.07 | 0.03±0.01 | 0.66±0.05 | -16.66±2.25 | 14.02±1.50 | 0.17±0.04 | 349.31±69.18 | 18.21±0.14 | 21.44±0.02 | 0.03±0.01 | 0.11±0.00 | 41.52±0.65 | 54.16±0.14 | 0.04±0.01 | 1.82±0.05 |
| LV (on-policy) | **-1.28**±0.06 | 15.11±1.36 | 0.23±0.06 | 144.02±77.54 | -2.56±0.27 | 77.96±13.03 | 0.42±0.09 | 2265.84±685.58 | **20.18**±0.02 | 21.43±0.02 | 0.04±0.01 | 0.06±0.00 | **49.10**±0.04 | 57.40±0.26 | 0.14±0.01 | 1.34±0.02 |
| LV + Buffer | -10.24±1.23 | 1.15±0.05 | 0.04±0.01 | 0.49±0.03 | -2.68±0.19 | 49.77±11.95 | 0.38±0.10 | 1566.19±596.76 | 19.40±0.03 | 21.03±0.01 | 0.03±0.00 | 0.08±0.00 | 44.83±0.41 | 54.29±0.08 | 0.05±0.01 | 1.83±0.07 |
| LV + Buffer + MCMC | -10.30±1.30 | 1.18±0.05 | 0.04±0.01 | 0.49±0.02 | -12.39±2.53 | 13.25±1.48 | 0.65±0.05 | 1070.42±499.38 | 19.30±0.08 | 21.03±0.01 | 0.03±0.01 | 0.08±0.00 | 41.80±0.22 | 54.52±0.07 | 0.08±0.01 | 2.23±0.03 |
| TB (on-policy) | -1.29±0.07 | 14.99±1.52 | 0.24±0.06 | 156.58±105.97 | -2.47±0.30 | 71.53±7.55 | 0.40±0.07 | 2142.65±637.28 | **20.18**±0.02 | 21.43±0.02 | 0.04±0.00 | 0.06±0.00 | 49.05±0.05 | 57.48±0.28 | 0.15±0.02 | 1.37±0.02 |
| TB + Buffer | -3.99±0.12 | **0.64**±0.01 | **0.02**±0.01 | 0.37±0.01 | -5.97±1.06 | 4.20±1.39 | 0.07±0.03 | 114.11±53.48 | 19.80±0.04 | **20.93**±0.01 | **0.02**±0.01 | 0.06±0.00 | 48.79±0.07 | 53.18±0.03 | **0.03**±0.01 | **1.30**±0.01 |
| TB + Buffer + MCMC | -3.84±0.15 | **0.64**±0.03 | 0.03±0.01 | **0.35**±0.02 | -7.13±0.73 | **0.91**±0.03 | **0.04**±0.01 | **4.25**±0.30 | 19.72±0.05 | **20.93**±0.01 | **0.02**±0.01 | 0.07±0.00 | 48.74±0.10 | **52.84**±0.02 | 0.04±0.01 | 1.36±0.02 |

| Ground Truth | MH | MDNS | TB (on-policy) | TB + Buffer | TB + Buffer + MCMC |

*Figure 3.* Visualisation of samples generated by each method for the discretised 40GMM ($d = 4 \times 8 = 32$), projected to the first two dimensions. The crosses are placed at the 40 modes of the mixture. Only the off-policy TB + Buffer + MCMC discovers all 40 modes.

challenging to approximate, even with algorithms that assume access to the energy gradient. See §C.2.1 for details.

The off-policy MCMC methods use Metropolis-Hastings MCMC with a 1-Hamming-ball proposal; see §C.1.2.

### 5.2.1. DIFFUSION SAMPLERS

We study the performance of discrete diffusion samplers with and without off-policy training methods. We use target distribution temperature annealing for all sampling methods and synthetic targets; we investigate the situation where no target annealing is applied in §D.1.1. For evaluation, we use the ELBO and EUBO metrics, as well as the sample-based Sinkhorn and MMD with RBF kernel. See §C.2 for details.

Table 2 shows performance metrics over all methods considered. The off-policy methods trained with the TB objective consistently outperform or match on-policy methods and other learning-based methods (LV, MDNS) in all metrics. Similar to what was seen in the Ising and Potts models, this improvement is most noticeable on challenging and highly multimodal problems such as the 4-dimensional 40GMM (32-dimensional binary code), where standard on-policy methods fail to effectively capture all modes of the distribution, as can be seen from Fig. 3 and Fig. 9 in §D.2.2.

We further use this task to study more closely the effect of off-policy training; see §D.1.2. We also use the 40GMM target to study the effect of different masking schedules in Table 10, and to study the effect of calculating gradients over a subset of the trajectory steps in Table 11.

### 5.2.2. SCHRÖDINGER BRIDGES

We investigate our algorithms' ability to construct discrete stochastic transport maps between various pairs of distributions in two dimensions, discretised to 16-dimensional binary Gray codes. In each pair, the first distribution is given by samples and the second by its energy. In Fig. 4 we show samples from the final bridges trained using IPF on three different problems. We compare performance of both on-policy and off-policy methods. While both methods succeed in fitting bridges between simple densities, the off-policy method scales better to the multimodal 40GMM target distribution, where the on-policy sampler collapses. See §C.3 for experimental details.

### 5.3. Outsourced Sampling

We show that discrete diffusion samplers can be used for posterior sampling in the latent spaces of generative models. We chose our generative model to be a VQ-VAE trained on MNIST (Deng, 2012) with 16-dimensional latent space and 8-word codebook ($\mathcal{S} = \{1, \ldots, 8\}^{16}$) using the algorithm in Hu et al. (2023), which allows joint learning of an autoregressive prior $p_{\text{latent}}(z)$ and decoder, whose predicted mean we treat as a deterministic decoding $x = f(z)$. In these experiments, the likelihood function $p(y \mid x)$ returns the probability that the generated image $x$ represents a digit class that is *odd*, *even*, or equal to some number.

In Fig. 5 we show images generated by our approach with both on-policy and off-policy training, showing that both are able to produce digits of the desired classes. Additional images, using different objectives and model architectures, are shown in §D.2.3.

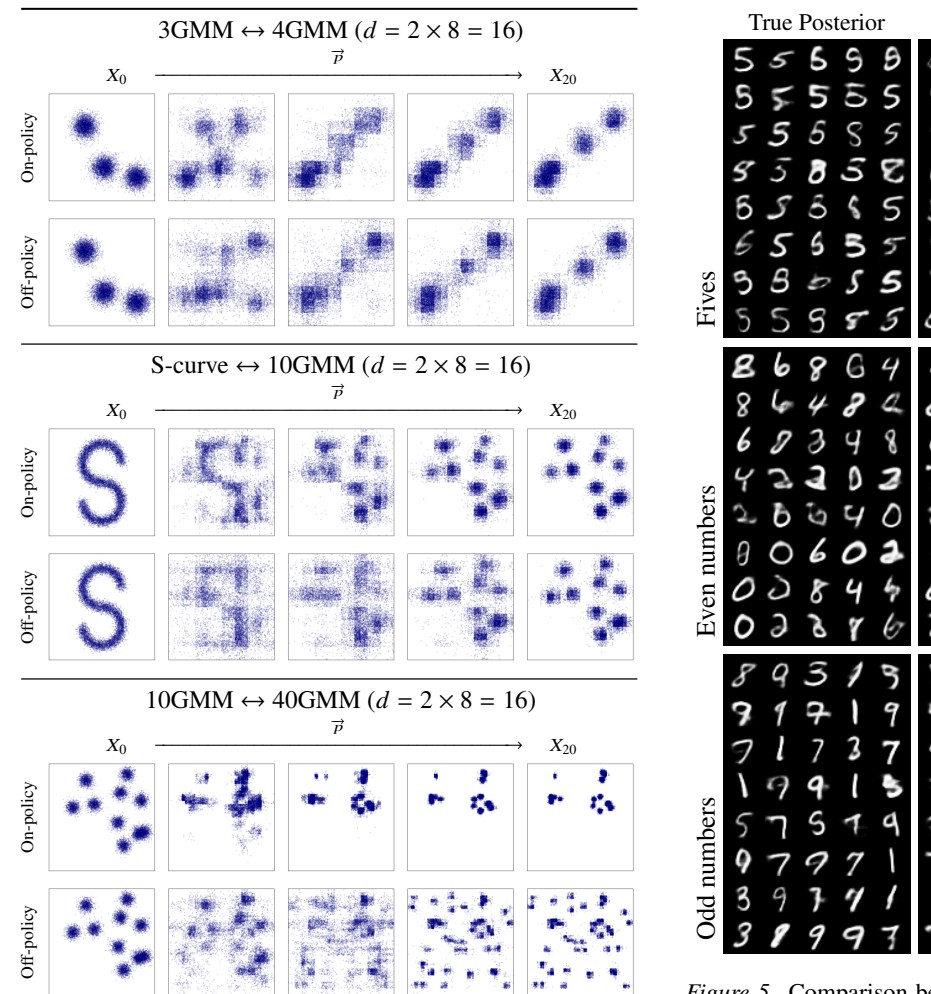

Figure 4. Comparison of discrete-space Schrödinger bridges on 16-dimensional binary Gray-coded spatial data learnt by data-to-energy IPF with on-policy and off-policy LV training.

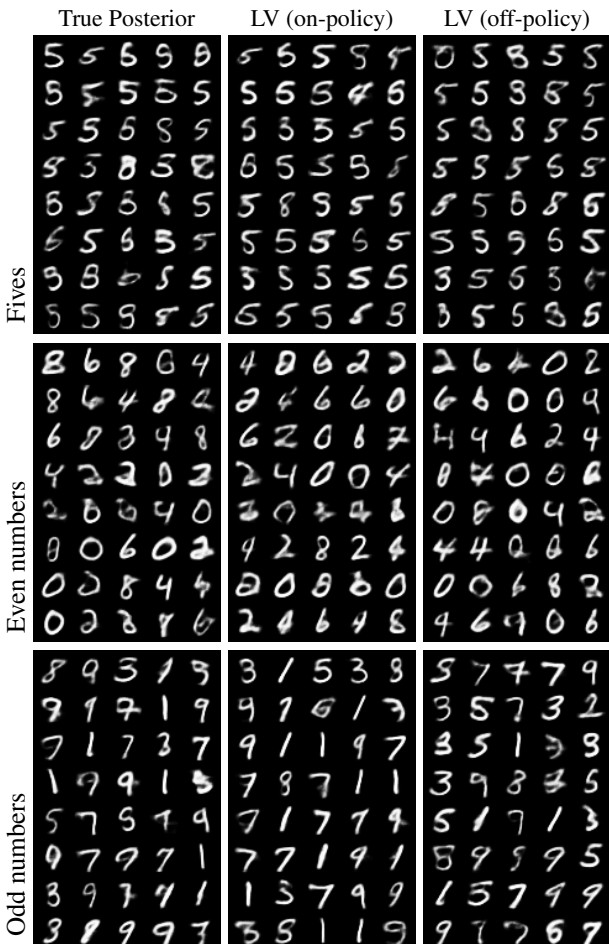

Figure 5. Comparison between decoded samples from the true posterior and from diffusion samplers (trained on- and off-policy) in the latent space of a VQ-VAE trained on MNIST, showing that outsourced discrete diffusion samplers successfully allow conditioning by learning to sample in latent space.

## 6. Related Work

**Amortised sampling.** Training a model to sample from some distribution given only by an unnormalised energy function is the setting of amortised sampling. Earlier neural amortised samplers in continuous space used normalising flows as the approximating families (Noé et al., 2019; Midgley et al., 2023). More recent methods fit a neural stochastic differential equation to transport some noise distribution to the target by attempting to learn the reversal of some noising process (*e.g.*, Zhang & Chen, 2022; Vargas et al., 2022; 2023; Richter et al., 2024; Akhound-Sadegh et al., 2024). Other approaches fit the process to a path of intermediate densities through time in a similar style to annealed importance sampling and sequential Monte Carlo (Neal, 2001; Del Moral et al., 2006), *e.g.*, Vargas et al. (2024); Albergo & Vanden-Eijnden (2025). Some methods represent hybrids of time-reversal and transport samplers or can

be augmented with sequential Monte Carlo (Chen et al., 2025a; Choi et al., 2026). Amortised sampling in discrete spaces has also been explored in Holderrieth et al. (2025), or in the setting of discrete diffusion samplers detailed in the next paragraph. Our work transfers methods from the continuous-space amortised sampling literature to discrete settings.

**Discrete diffusion samplers.** Discrete diffusion has gained recent traction for many applications (*inter alia* Campbell et al., 2022; Avdeyev et al., 2023; Sahoo et al., 2024), or, in the discrete diffusion sampling case, for applications such as statistical physics and combinatorial optimisation (Sanokowski et al., 2025a; 2024). Zhu et al. (2025) formulated the problem using stochastic optimal control with a masked diffusion framework that is equivalent to any-order autoregressive modelling; this work was later extended by Guo et al. (2026). The setting of single unmasking

considered by MDNS is equivalent to the setting of Zhang et al. (2022a), which described the problem in terms of generative flow networks – a general family of off-policy RL algorithms algorithms for amortised sampling by sequential decision-making originally developed in discrete space (Bengio et al., 2021; 2023) but generalised to continuous spaces (Lahlou et al., 2023) and used for diffusion-based sampling. The algorithms we use in this paper for discrete diffusion samplers are directly connected with past uses of these algorithms for continuous-space diffusion (Sendera et al., 2024; Choi et al., 2026; Tiapkin et al., 2024; Shen et al., 2023).

**Schrödinger bridges.** Initially defined by Schrödinger (1931; 1932), Schrödinger bridges have been explored in the continuous state space setting (De Bortoli et al., 2021; Vargas et al., 2021; Chen et al., 2021; 2022; Shi et al., 2023; Tong et al., 2024) and the discrete state space setting (Kim et al., 2025a; Ksenofontov & Korotin, 2025). Recently, the iterative proportional fitting algorithm (IPF; Fortet, 1940; Kullback, 1968; Chen et al., 2021) was adapted by Tamogashev & Malkin (2026) to be able to solve the Schrödinger bridge problem in the data-to-energy and energy-to-energy settings where at least one side of the bridge is given only by an unnormalised density function. We extend this data-to-energy algorithm into discrete spaces with a uniform diffusion reference process.

## 7. Conclusion

We have shown that using off-policy reinforcement learning methods in training of discrete diffusion samplers can significantly improve their performance. We have demonstrated this in our evaluation on previously considered densities and on more challenging discrete problems, in which mode collapse is a concern. We have illustrated, for the first time, discrete Schrödinger bridges in the case where one distribution is given only by an energy function and have further tested the capabilities of discrete diffusion samplers on the novel application of posterior sampling in discrete latent spaces.

The methods studied here have certain limitations. Some of these limitations are inherited from the discrete diffusion sampling framework itself, such as the need to fix a noising kernel (which can be alleviated by bridge models or otherwise learning the noising kernel, as was recently shown to be beneficial in continuous spaces by Gritsaev et al. (2025); Sanokowski et al. (2025b)). Others are related to the computational cost of rolling out and storing trajectories; our finding that discrete diffusion samplers can be trained at coarser time scales than those at which they are sampled (and thus at lower cost) begs for an extension of Berner et al. (2026)'s theory regarding the continuous-time limit of diffusion samplers to the discrete-space case.

Beyond the resolution of these limitations, we can identify three practical directions for future work. First, the capabilities of discrete diffusion samplers could be evaluated on problems involving learning an energy function from a dataset (as studied by Zhang et al. (2022a) using methods that are a special case of those we consider). Second, while our work has made a step in improving discrete diffusion samplers, they have yet to be scaled to large problems, such as outsourced sampling for large image generative models, where continuous-space diffusion samplers have been successful (Venkatraman et al., 2025). Third, the potential and applications of Schrödinger bridge algorithms, particularly in latent spaces, for data-free alignment of distributions have yet to be fully understood.

## Acknowledgements

The work of AC is supported by the UKRI EPSRC through the CDT in Machine Learning Systems hosted in the School of Informatics, University of Edinburgh (EP/Y03516X/1), as well as joint sponsorship from Level E Research. The work of SC, VE, and ESW is supported by the Advanced Research and Invention Agency (ARIA). ESW acknowledges support from the CIFAR Learning in Machines and Brains programme.

This work was enabled by the computational resources of the Edinburgh International Data Facility (EIDF) and Edinburgh Generative AI Laboratory (GAIL).

## Impact Statement

This paper presents work whose goal is to advance the field of machine learning. There are many potential societal consequences of our work, none of which we feel must be specifically highlighted here.

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

# A. Noising Kernels for Discrete Diffusion Samplers

## A.1. Masked Diffusion Sampling

Masked diffusion models (Austin et al., 2021) have being successfully applied to a variety of problems, including language generation (Shi et al., 2024; Sahoo et al., 2024), protein design (Gruver et al., 2023) and graph generation (Vignac et al., 2023). Recent work has shown the stochastic optimal control formulation of masked diffusion in the problem of discrete state space sampling (Zhu et al., 2025), though only the autoregressive single-unmasking case was considered.

Masked diffusion sampling involves appending to our state space a mask token that we denote with $\star$. The non-terminal states in a trajectory then take values in $\overline{\mathcal{S}} = \{\{1, \ldots, C\} \cup \{\star\}\}^d$, with the trajectory value at time $N$ still taking values in $\mathcal{S}$ in order to be able to satisfy the requirement that the terminal marginal equals the target distribution.

Masked diffusion samplers define the initial distribution to be the Dirac delta distribution over the fully masked state $p_0 = \delta_{(\star, \ldots, \star)}$, and define the reverse process as the masking process, where at step $n + 1$ we randomly choose $k_{n+1}$-elements from the set of elements in $X_{n+1}$ not equal to $\star$, and replace them with $\star$.

Define the set of non-mask elements at time $n$ as $\bullet_n = \{i : (X_n)_i \neq \star\}$, following the notational convention of Zhu et al. (2025) we use $X^{I \leftarrow V}$ to denote the operation of replacing the values of all elements of $X$ from some index set $I = \{i_1, i_2, \ldots, i_k\}$, where $i_j \in \{1, \ldots, d\}$, with those from the set $V = \{v_1, v_2, \ldots, v_k\}$ where $v_j \in \overline{\mathcal{S}}$. In equations:

$$(X^{I \leftarrow V})_j = \begin{cases} X_j & j \notin I \\ v_j & j \in I \end{cases}, V = \{v_1, \ldots, v_{|I|}\}.$$

$$\overleftarrow{p}(X_n \mid X_{n+1}) = \frac{1}{\binom{|\bullet_{n+1}|}{k_{n+1}}} \mathbf{1}_{X_n = X_{n+1}^{I \leftarrow \{\star, \ldots, \star\}}}$$

To learn the forward process we parameterise the forward kernels using a neural network that takes in $X_n \in \overline{\mathcal{S}}$ and outputs a matrix of size $d \times |\mathcal{S}|$ of probabilities summing to 1 along each row – element $(i, j)$ is the probability of element $i$ taking value $j$ at the next step given that element $i$ has been chosen to be unmasked – we then randomly choose the $k_{n+1}$ elements of $X_n$ to be unmasked and sample the corresponding elements of $X_{n+1}$ from the appropriate categorical distribution, keeping all other values the same.

$$\overrightarrow{p}_\theta(X_{n+1} = X_n^{I \leftarrow V} \mid X_n) = \frac{1}{\binom{d - |\bullet_n|}{k_{n+1}}} \prod_{\substack{i \in I \\ v \in V}} f_\theta(X_n)_{i,v} \mathbf{1}_{(X_n)_i = \star}$$

The set $\{k_1, \ldots, k_N\}$ we refer to as the masking schedule; its choice can greatly affect performance. We require that $\sum_{n=1}^N k_n = d$ so that we go from $X_N \in \mathcal{S}$ to the fully masked state at $X_0$. Smaller values for the $k_n$ or, correspondingly, a larger $N$ typically yield more accurate results when attempting to model the conditional distributions. Picking all $k_n = 1$, and therefore $N = d$, corresponds to the any-order autoregressive single-unmasking case which can be seen as exact in the sense that there is no error introduced by the process of unmasking multiple, potentially correlated, elements simultaneously. The trade-off here is in terms of algorithmic speed, lower $N$ means fewer function evaluations and gradient calculations. In our experiments we make use of a random masking schedule where at each backward (forward) step we mask (unmask) randomly between $K_{\min}$ and $K_{\max}$ elements.

## A.2. Uniform Diffusion Sampling

Uniform diffusion sampling (Austin et al., 2021) allows us to stay within the target state space $\mathcal{S}$ at all points along the trajectory. This allows its use in the Schrödinger Bridge problem where we bridge between two distributions.

**Binary setting.**  First, for simplicity, we constrain ourselves to the binary state-space setting $\mathcal{S} = \{0, 1\}^d$. Uniform diffusion sampling selects the initial distribution, $p_0$, to be the uniform distribution over $\mathcal{S}$. In the binary setting the backward process is simply a flipping procedure where each element of the current state is flipped with some probability $p_{\text{flip}}$.

$$\overleftarrow{p}(X_n \mid X_{n+1}) = \prod_{i=1}^d p_{\text{flip}}^{|(X_n)_i - (X_{n+1})_i|}.$$

The forward process can then simply be parameterised using a neural network, $f_\theta$, to predict a $d$-dimensional vector of flipping probabilities.

$$\overrightarrow{p}_\theta(X_{n+1} \mid X_n) = \prod_{i=1}^{d} f_\theta(X_n)_i^{|(X_n)_i - (X_{n+1})_i|}$$

Unlike masked diffusion which guarantees that the noising process ends at the initial distribution, in uniform discrete diffusion sampling it is crucial that appropriately large values for $N$ or $p_{\text{flip}}$ are taken such that $X_0$ is distributed uniformly over $\mathcal{S}$.

**Non-binary setting.** To extend the previous definition into the state space $\mathcal{S} = \{1, \ldots, C\}^d$ we again define the $p_0$ as the uniform distribution over $\mathcal{S}$. Similarly to the binary setting we also decide whether to change the value of a particular element of the state with some probability $p_{\text{flip}}$, however if we decide to flip some element then it can take $C - 1$ possible values and so we take the uniform distribution over these.

$$\log \overleftarrow{p}(X_n \mid X_{n+1}) = \sum_{i=1}^{d} \mathbf{1}_{(X_n)_i = (X_{n+1})_i} \log(1 - p_{\text{flip}}) + (\mathbf{1}_{(X_n)_i \neq (X_{n+1})_i}) \log \frac{p_{\text{flip}}}{C - 1}$$

The forward process can then be parameterised by a neural network $f_\theta$ which outputs a matrix of probabilities corresponding to the next step categorical distributions for each element of the current state, ensuring that the rows of the matrix sum to 1.

## B. Algorithms

In this section we detail two algorithms we use for training. Algorithm 1 is used for Ising and Potts models, for synthetics tasks, and for outsourced experiments on MNIST. Algorithm 2 is used to learn discrete data-to-energy Schrödinger bridge. Note, that Algorithm 2 details both off-policy and on-policy versions, with modifications required for off-policy training being highlighted in blue.

---

**Algorithm 1** Off-policy training with MCMC exploration

---

**Require:** Target energy $\mathcal{E}$, initial distribution $p_0$, forward and backward kernels $\overrightarrow{p}_\theta$ and $\overleftarrow{p}$, number of epochs $N_{\text{epoch}}$, batch size $M$, buffers $\mathcal{B}$ and $\mathcal{B}_{\text{MCMC}}$, off-to-on ratio $R$, MCMC interval $I$, MCMC num steps $L$, MCMC sample ratio $r$.

1: **for** $i = 1, \ldots, N_{\text{epoch}}$ **do**
2:     **if** $(i - 1) \bmod R = 0$ **then**      ▷ On-policy sampling
3:         Sample trajectories $\{x_{0:N}^m\}_{m=1}^M$ from $p_0 \otimes \overrightarrow{p}_\theta^{\otimes N}$ and compute $w^m$ for each $x_{0:N}^m$.      ▷ Eq. (4)
4:         Update buffer, $\mathcal{B} \leftarrow \mathcal{B} \cup \{(x_N^m, w^m)\}_{m=1}^M$.
5:     **else**      ▷ Off-policy sampling
6:         Let $M' = \min(\lceil rM \rceil, |\mathcal{B}_{\text{MCMC}}|)$
7:         Sample $\{x^m\}_{m=1}^{M'}$ from $\mathcal{B}_{\text{MCMC}}$ uniformly, and sample $\{x^{M'+m}\}_{m=1}^{M-M'}$ from $\mathcal{B}$ proportionally to $w^m$.
8:         From each $x^m$, sample backward trajectories $\{x_{0:N}^m\}_{m=1}^M$ using $\overleftarrow{p}^{\otimes N}$.
9:     **end if**
10:    Update $\theta$ using $\nabla_\theta \mathcal{L}^{\mathbb{P}}$ given $\{x_{0:N}^m\}_{m=1}^M$.      ▷ Eq. (3)
11:    **if** $(i - 1) \bmod I = 0$ **then**      ▷ MCMC exploration
12:        Sample $\{x^m\}_{m=1}^M$ from $\mathcal{B}$ proportionally to $w^m$.
13:        Run $M$ parallel MCMC chains for $L$ steps, each starting from $x^m$ and take the final state of each chain, $\{x_L^m\}_{m=1}^M$.
14:        Update MCMC buffer, $\mathcal{B}_{\text{MCMC}} \leftarrow \mathcal{B}_{\text{MCMC}} \cup \{x_L^m\}_{m=1}^M$.
15:    **end if**
16: **end for**

---

---

**Algorithm 2** Training of data-to-energy discrete Schödinger bridges. Parts used for off-policy training are marked by blue.

---

**Require:** Distribution $p_0$, distribution $p_N$ with queryable density, backward model $\overleftarrow{p}_\varphi$, forward model $\overrightarrow{p}_\theta$ number of epochs $N_{\text{sb\_iter}}$, number of trajectories for forward model training $K_{\text{traj}}$, buffers $\mathcal{B}$ and $\mathcal{B}_{\text{MCMC}}$, MCMC num steps $L$.

1: **for** $sb\_iter = 1, \ldots, N_{\text{sb\_iter}}$ **do**
2:     **while** *not converged* **do**                           ▷ Train backward model to maximise log likelihood.
3:         Sample $\tau = x_{0:N} \sim p_0 \otimes \overrightarrow{p}_\theta^{\otimes N}$;
4:         Update $\varphi$ using $\nabla_\varphi \log \overleftarrow{p}_\varphi(\tau \mid x_N)$.
5:     **end while**
6:     **if** *use off-policy training* **then**
7:         Sample a batch of $x_N \sim \mathcal{B}$;
8:         Run MCMC for $L$ steps on $x_N$, obtaining $x'_N$;
9:         Update buffer $\mathcal{B}_{\text{MCMC}}$ with samples $x'_N$.
10:     **end if**
11:     **while** *not converged* **do**                       ▷ Train forward model using Log Variance divergence.
12:         **if** *on-policy* **then**
13:             $x_0 \sim p_0(x)$, $\tau^{(j)} \sim \overrightarrow{p}_\theta^{\otimes N}(\tau|x_0)$, $\tau = x_{0:N}$ for $1 \le j \le K_{\text{traj}}$;
14:             Update buffer $\mathcal{B}$ with samples $x_N$.
15:         **else**
16:             $x_N \sim \mathcal{B}_{\text{MCMC}}$, $\tau^{(1)} \sim \overleftarrow{p}_\varphi^{\otimes N}(\tau \mid x_N)$, $\tau^{(1)} = x_{0:N}$;
17:             $\tau^{(j)} \sim \overrightarrow{p}_\theta^{\otimes N}(\tau|x_0)$ for $2 \le j \le K_{\text{traj}}$.
18:         **end if**
19:         Update $\theta$ using $\nabla_\theta \text{Var}_j \left( \log \frac{\overrightarrow{p}_\theta^{\otimes N}(\tau^{(j)}|x_0^{(j)})}{\overleftarrow{p}_\varphi^{\otimes N}(\tau^{(j)}|x_N^{(j)})} - \log p_N(x_N^{(j)}) \right)$
20:     **end while**
21: **end for**

---

## C. Experimental Details

### C.1. Ising and Potts models

#### C.1.1. TARGETS

Full detail is given in the main text, see (7) for the Potts energy and the following discussion for the special case of the Ising model.

#### C.1.2. BASELINES

We consider Metropolis-Hastings with $H$-Hamming-ball proposals and Masked Diffusion Neural Sampler (Zhu et al., 2025), which shares the same sampling process as ours, as baselines for benchmarking on Ising and Potts models. LEAPS (Holderrieth et al., 2025) is another seminal work in discrete diffusion for sampling, but we exclude it from benchmarking due to differences in the sampling process.

**Metropolis-Hastings with $H$-Hamming-ball proposal (MH).** $H$-Hamming-ball MH employs a proposal kernel that samples candidates within a Hamming distance $H$ from the current state. Specifically, a 1-Hamming-ball proposal uniformly chooses an index $i \in \{1, \ldots, d\}$ and replaces $x_i$ with a new value drawn uniformly from $\{1, \ldots, C\} \setminus \{x_i\}$. The $H$-Hamming-ball proposal applies the 1-Hamming-ball proposal $H$ times sequentially before the Metropolis-Hastings acceptance step.

We use the above $H$-Hamming-ball MH algorithm as a baseline for the Ising and Potts models. We tuned $H$ over $\{1, 2, 5, 10\}$ and found $H = 1$ to be the best setting across all Ising and Potts configurations we considered. We ran 128 chains in parallel, with each chain running 2000 steps for burn-in followed by 25600 steps to collect 128 samples (1 sample every 200 steps), resulting in $128 \times 128 = 16384$ samples.

**Masked Diffusion Neural Sampler (MDNS).** We adopt the same architecture as in the original MDNS paper, a vision transformer with rotary positional embedding (Heo et al., 2024). We use 2 blocks with 64-dimensional embeddings and 4 attention heads for the Ising model, while we increase the embedding dimension to 128 for the Potts model. Note that the model we used is smaller than the one in the original MDNS paper (4 or 6 blocks), but we found that the smaller model is

sufficient for our purpose.

We use weighted denoising cross-entropy (WDCE) loss for MDNS, which is the default option for Ising and Potts models in their paper (see Section 3.3 Zhu et al. (2025)). Since their algorithm with WDCE reuses a batch of samples for multiple gradient steps, we set the batch size as 256 and the number of epochs as 50000, ensuring the number of function evaluations is comparable to ours. Additionally, while the MDNS paper utilised a "warm-up" phase involving pre-training with a tempered target for a fixed number of epochs, we adopt the same annealing schedule as in our methods (see §D.1.1), which we found to be more effective.

### C.1.3. EVALUATION METRICS

This section introduces evaluation metrics for discrete diffusion samplers. We use three target-agnostic evaluation metrics – evidence lower bound (ELBO), evidence upper bound (EUBO; Blessing et al., 2024), and Sinkhorn – as well as two target-specific metrics – magnetisation error and 2-point correlation error. For all metrics, we use $M_{\text{eval}} = 16384$ evaluation samples. Note that EUBO, Sinkhorn, and 2-point correlation error require samples from target distributions, which we obtained from a sufficiently long run of the Swendsen-Wang algorithm; $2^{16}$ burn-in steps followed by $2^{24}$ steps (collect a sample every $2^{10}$ steps).

**ELBO.** Consider the reverse KL divergence between $p_0 \otimes \overrightarrow{p}_\theta^{\otimes N}$ and $p_{\text{target}} \otimes \overleftarrow{p}^{\otimes N}$:

$$\text{KL}\left(p_0 \otimes \overrightarrow{p}_\theta^{\otimes N} \,\|\, p_{\text{target}} \otimes \overleftarrow{p}^{\otimes N}\right) = \underset{X_{0:N} \sim p_0 \otimes \overrightarrow{p}_\theta^{\otimes N}}{\mathbb{E}}\left[\log \frac{p_0 \otimes \overrightarrow{p}_\theta^{\otimes N}(X_{0:N})}{p_{\text{target}} \otimes \overleftarrow{p}^{\otimes N}(X_{0:N})}\right] = -\underbrace{\underset{X_{0:N} \sim p_0 \otimes \overrightarrow{p}_\theta^{\otimes N}}{\mathbb{E}}\left[\log w\right]}_{=:\text{ELBO}} + \log Z,$$

where $w$ is defined in (4). Due to the non-negativity of the KL divergence, ELBO gives a lower bound on $\log Z$. We approximate ELBO by $M_{\text{eval}}$ samples.

**EUBO.** Similar to ELBO, consider the forward KL divergence between $p_{\text{target}} \otimes \overleftarrow{p}^{\otimes N}$ and $p_0 \otimes \overrightarrow{p}_\theta^{\otimes N}$:

$$\text{KL}\left(p_{\text{target}} \otimes \overleftarrow{p}^{\otimes N} \,\|\, p_0 \otimes \overrightarrow{p}_\theta^{\otimes N}\right) = \underset{X_{0:N} \sim p_{\text{target}} \otimes \overleftarrow{p}^{\otimes N}}{\mathbb{E}}\left[\log \frac{p_{\text{target}} \otimes \overleftarrow{p}^{\otimes N}(X_{0:N})}{p_0 \otimes \overrightarrow{p}_\theta^{\otimes N}(X_{0:N})}\right] = \underbrace{\underset{X_{0:N} \sim p_{\text{target}} \otimes \overleftarrow{p}^{\otimes N}}{\mathbb{E}}\left[\log w\right]}_{=:\text{EUBO}} - \log Z.$$

EUBO gives an upper bound on $\log Z$. Note that EUBO requires true samples from $p_{\text{target}}$. Again, we use a Monte Carlo estimate with $M_{\text{eval}}$ samples. Blessing et al. (2024) suggested EUBO as a metric for measuring mode coverage.

**Sinkhorn distance.** The Sinkhorn distance (Cuturi, 2013) measures the entropy-regularised optimal transport cost between two distributions. We employ the Hamming distance as a distance metric and set the regularisation parameter to 0.001. We compute the distance using two sets of $M_{\text{eval}}$ samples, drawn from the target distribution and our model $\overrightarrow{p}$, respectively.

**Ising Magnetisation.** The state space for the Ising model is $\mathcal{S} = \{-1, 1\}^{L \times L}$. We compute the metrics with respect to some sampling distribution of the spins, $\pi$. The magnetisation of a spin $i$ is defined as $M(i) = \mathbb{E}_{x \sim \pi}[x_i]$. We define the average magnetisation of the rows and columns as

$$M^{\text{row}}(k) = \frac{1}{L} \sum_{i \in \text{row}(k)} M(i), \qquad M^{\text{col}}(k) = \frac{1}{L} \sum_{i \in \text{col}(k)} M(i),$$

where $\text{row}(k)$ and $\text{col}(k)$ denote sets of indices corresponding to the spins in row $k$ and column $k$ respectively. We then define the magnetisation error as

$$M_{\text{error}} = \frac{1}{2L} \sum_{k=1}^{L} |M^{\text{row}}(k) - M_{\text{true}}^{\text{row}}(k)| + |M^{\text{col}}(k) - M_{\text{true}}^{\text{col}}(k)|$$

**Ising 2-point correlation.** The 2-point correlation of two spins, $i, j$ on the lattice is given by $C(i,j) = \mathbb{E}_{x \sim \pi}[x_i x_j] - \mathbb{E}_{x \sim \pi}[x_i]\mathbb{E}_{x \sim \pi}[x_j]$. In a similar fashion to the above row magnetisation, we average all row/column correlations at a distance

$r$ along the corresponding dimension,

$$C^{\text{row}}(r) = \frac{1}{L^2} \sum_{k=1}^{L} \sum_{\substack{i \in \text{row}(k) \\ j \in \text{row}((k+r) \mod L) \\ i,j \text{ same column}}} C(i,j), \qquad C^{\text{col}}(r) = \frac{1}{L^2} \sum_{k=1}^{L} \sum_{\substack{i \in \text{col}(k) \\ j \in \text{col}((k+r) \mod L) \\ i,j \text{ same row}}} C(i,j).$$

We then define the two-point correlation error as

$$\frac{1}{2L} \sum_{r=0}^{L-1} |C^{\text{row}}(r) - C^{\text{row}}_{\text{true}}(r)| + |C^{\text{col}}(r) - C^{\text{col}}_{\text{true}}(r)|.$$

**Potts magnetisation.** Recall the Potts model state space $\mathcal{S} = \{1, \ldots, q\}^{L \times L}$. The Potts magnetisation calculated for a batch $B$ of samples is defined as

$$M(i) = \frac{q \max_{1 \le c \le q} (\mu_c^i / |B|) - 1}{q - 1},$$

where $\mu_c^i = \sum_{x \in B} \mathbf{1}_{x_i = c}$ is the number of samples in the batch for which spin $i$ has value $c$. The values for row/column magnetisation and the magnetisation error are identical to those described in the Ising model, but with the above definition of magnetisation substituted in.

**Potts 2-point correlation.** The 2-point correlation of the Potts model is defined as

$$C(i,j) = \mathbb{E}\left[\mathbf{1}_{x_i = x_j} - \frac{1}{q}\right].$$

Again, the definitions of average row/column correlation and 2-point correlation error are identical to the Ising model but with the above definition of 2-point correlation substituted in.

### C.1.4. ALGORITHM DETAILS, MODEL ARCHITECTURES, AND HYPERPARAMETERS

For all learning-based methods, we use the same ViT architecture with a rotary position embedding as in MDNS, which is described in §C.1.2, and AdamW (Loshchilov & Hutter, 2019) as optimiser. We anneal the target temperature during training to improve training; see §D.1.1 for the ablation study. For TB and LV, we employ multiple unmasking (see the last paragraph of §2.1, §A.1, and results in §D.1.4) with $K_{\min} = K_{\max} = 4$, *i.e.*, our forward kernel $\overrightarrow{p}$ unmasks 4 elements at each step, while we use single-step unmasking for evaluation. We use single unmasking for MDNS. For off-policy training, we use the importance-weighted buffer as described in §2.2, and the Swendsen-Wang algorithm (Swendsen & Wang, 1986; 1987) for MCMC exploration. Table 3 lists the hyperparameters used for the Ising and Potts experiments.

*Table 3.* Hyperparameter settings for Ising and Potts model experiments.

|  | Ising ($L = 16$) | Potts ($L = 16$) |
|---|---|---|
| Model type | | ViT |
| Number of blocks | | 2 |
| Number of attention heads | | 4 |
| Embedding dim | 64 | 128 |
| Learning rate | | 1e-3 |
| Number of training epochs ($N_{\text{epochs}}$) | | 20000 |
| Batch size ($M$) | | 128 |
| Buffer size | | 12800 |
| Off-to-On ratio ($R$) | | 2 |
| MCMC sample ratio ($r$) | | 0.2 |
| MCMC interval ($I$) | | 500 |
| MCMC steps (incl. burn-in) ($L$) | | 100 |

## C.2. Synthetic Targets

### C.2.1. TARGETS

We define the discrete synthetic target densities as discretisations of continuous distributions using Gray codes (Gray, 1953).

*Table 4.* Conversion from decimal to Gray code ($b = 3$).

| Decimal | Gray |
|---------|------|
| 0 | 000 |
| 1 | 001 |
| 2 | 011 |
| 3 | 010 |
| 4 | 110 |
| 5 | 111 |
| 6 | 101 |
| 7 | 100 |

**Gray-code discretisation.** To discretise a $D$-dimensional continuous space, we first truncate each dimension to the interval $[-R, R]$, then shift by $R$ and scale by $\frac{1}{2R}$ to map them to $[0, 1]^D$. Each interval is then quantised into integers $\{0, \ldots, 2^b - 1\}$, where $b$ represents the number of quantisation bits. The specific parameters for each target are provided below. In order to transform each bin into a (higher-dimensional) binary variable, we use Gray codes. Gray code orders binary numbers in such a way that two values which are consecutive in the original discretised space only differ by a single bit in Gray code. An example is given in Table 4. For $D$-dimensional space with $b$ quantisation bits, the dimension of the resulting discrete space is $d = D * b$.

Given the energy function $\widetilde{\mathcal{E}}$ defined in $\mathbb{R}^D$, we define the energy $\mathcal{E}$ in gray-coded discrete space $\{0, 1\}^d$ as follows:

$$\mathcal{E}(x) = \widetilde{\mathcal{E}}\left((\texttt{InverseGray}(x) + 0.5) * 2R - R\right) - D \log \frac{R}{2^{b-1}},$$

where $x \in \{0, 1\}^d$ is a binary vector, $\texttt{InverseGray}$ is the inverse mapping of Gray code. We add 0.5 to shift the integer values from $\texttt{InverseGray}$ and undo the scaling and shifting. The last term is to account for the size of the bin.

We consider two targets, *40GMM* and *ManyWell*, which are common synthetic benchmarks for continuous sampling:

- **40GMM** ($D \in \{2, 4\} \mid R = 50 \mid b = 8$) (Midgley et al., 2023) is a uniform mixture of 40 Gaussians in $\mathbb{R}^D$, where each mixture component has mean $c_i \in \mathbb{R}^D$ and identity covariance $I$. Each dimension of the $c_i$ are sampled from the uniform distribution $\mathcal{U}[a, b]$ where the upper and lower bounds were chosen so that the resulting Gaussian mixture has small probability density outwith the truncation region, $a = -R + 3 = -47$ and $b = -R + S - 3 = 47$. The energy function of the 40GMM in the continuous space is given by:

$$\widetilde{\mathcal{E}}_{\text{40GMM}}(\tilde{x}) = -\frac{1}{40} \sum_{i=1}^{40} \log p_{\mathcal{N}}(\tilde{x}; c_i, I),$$

where $\tilde{x} \in \mathbb{R}^D$ and $p_{\mathcal{N}}(\cdot; c, I)$ is the density function of the normal distribution with mean $c$ and covariance $I$.

- **ManyWell** ($D \in \{4, 10\} \mid R = 4 \mid b = 8$) (Noé et al., 2019; Nüsken & Richter, 2021) is a product of $D/2$ identical DoubleWell distributions and $D/2$ standard Gaussian distributions. The DoubleWell energy is given by:

$$\widetilde{\mathcal{E}}_{\text{DoubleWell}}(\tilde{x}_1) = \tilde{x}_1^4 - 6\tilde{x}_1^2 - 0.5\tilde{x}_1.$$

where $\tilde{x}_1 \in \mathbb{R}$. Then, the ManyWell energy is defined as:

$$\widetilde{\mathcal{E}}_{\text{ManyWell}}(\tilde{x}) = \sum_{i=1}^{D/2} \widetilde{\mathcal{E}}_{\text{DoubleWell}}(\tilde{x}_{2i}) + \frac{1}{2}\tilde{x}_{2i+1}^2,$$

where $\tilde{x} \in \mathbb{R}^D$. We use rejection sampling to generate ground-truth samples.

To make the problem more challenging, we consider a variation called "Rotated" ManyWell, where we rotate each pair of dimensions $(x_{2i}, x_{2i+1})$ by $\pi/4$, introducing dependencies between them. The energy function of the Rotated ManyWell is defined as

$$\widetilde{\mathcal{E}}_{\text{RotatedManyWell}}(\tilde{x}) = \sum_{i=1}^{D/2} \widetilde{\mathcal{E}}_{\text{DoubleWell}}\left(\frac{\tilde{x}_{2i} + \tilde{x}_{2i+1}}{\sqrt{2}}\right) + \frac{-\tilde{x}_{2i} + \tilde{x}_{2i+1}}{2\sqrt{2}}.$$

We adopt the Rotated ManyWell as the default benchmark throughout the paper.

### C.2.2. BASELINES

We consider the same baselines as our experiments on the Ising and Potts models, which are detailed in §C.1.2. For the MH baseline, we use the same hyperparameters as described in §C.1.2, except that we tune $H$ of the Hamming-ball proposal over $\{1, 2, 5, 10\}$ and choose to use $H = 5$, which yields the best average rank of MMD over the four benchmarks that we consider. For MDNS, we use the same MLP architecture as our LV or TB algorithms, while using a larger batch size (256) and a larger number of epochs (50000) to make the number of function evaluations comparable to those of our algorithms.

### C.2.3. EVALUATION METRICS

For synthetic targets, we also use ELBO, EUBO, and the Sinkhorn distance, as explained in §C.1.3, whereas we measure the Sinkhorn distance in the continuous space using the $L_2$ distance instead of the Hamming distance. In addition to those, we compute the maximum mean discrepancy (MMD) using a Gaussian kernel, the $L_2$ distance, and the median heuristic (Gretton et al., 2012), also with $M_{\text{eval}} = 16384$ samples from the target and the model.

### C.2.4. ALGORITHM DETAILS, MODEL ARCHITECTURES, AND HYPERPARAMETERS

For all learning-based methods, including our TB and LV algorithms and also MDNS, we use a simple multi-layer perceptron (MLP) with a size chosen to be appropriate for the difficulty of the problem, with the AdamW optimiser (Loshchilov & Hutter, 2019). Temperature-annealed training is applied by default; see §D.1.1 for the ablation. For TB and LV algorithms, we use multiple unmasking only for training ManyWell 10D, with $K_{\text{min}} = K_{\text{max}} = 4$. We use the Metropolis-Hastings with $H$-Hamming-ball proposal (§C.1.2) with $H = 5$ as our off-policy MCMC exploration method, along with the importance-weighted buffer. Table 5 lists a set of hyperparameters of our algorithm that we use for the synthetic targets.

*Table 5.* Hyperparameter settings for experiments with synthetic targets.

| | 40GMM ($d = 2 \times 8 = 16$) | 40GMM ($d = 4 \times 8 = 32$) | ManyWell ($d = 4 \times 8 = 32$) | ManyWell ($d = 10 \times 8 = 80$) |
|---|---|---|---|---|
| Model type | | | MLP | |
| Number of layers | 2 | 4 | 2 | 4 |
| Hidden dim | | | 256 | |
| Learning Rate | | | 1e-3 | |
| Number of training epochs ($N_{\text{epochs}}$) | | | 20000 | |
| Batch size ($M$) | | | 128 | |
| Buffer size | | | 12800 | |
| Off-to-On ratio ($R$) | | | 2 | |
| MCMC sample ratio ($r$) | | | 0.2 | |
| MCMC Interval ($I$) | | | 500 | |
| MCMC Steps (incl. burn in) ($L$) | | | 100 | |

### C.3. Data-to-energy Schrödinger bridges

We conduct experiments on three pairs of distributions: 3GMM and 4GMM, S-Curve and 10GMM, 10GMM and 40GMM. All distributions have spatial dimension $D = 2$, which is discretised using 8-bit Gray code. We use the same neural network for parametrisation of both forward and backward processes. Hyperparameters are given in Table 6.

### C.4. Outsourced Sampling

#### C.4.1. ALGORITHM
#### DETAILS, MODEL ARCHITECTURES, AND HYPERPARAMETERS

We sample the latent space of a VQ-VAE model, that is trained following the method provided in Hu et al. (2023). The latent space has

*Table 6.* Hyperparameter settings for Schrödinger bridge experiments with synthetic targets.

| Model type | MLP |
|---|---|
| Number of layers | 4 |
| Hidden dim | 256 |
| Forward process learning rate | 1e-4 |
| Backward process learning rate | 1e-3 |
| Number of outer SB steps (num_sb_iter) | 50 |
| Number of forward steps (num_fwd_iter) | 2500 |
| Number of backward steps (num_bwd_iter) | 2500 |
| Batch size ($M$) | 256 |
| Buffer size | 25600 |
| Off-to-On ratio ($R$) | 10 |
| MCMC sample ratio ($r$) | 0.5 |
| MCMC Steps (incl. burn in) ($L$) | 200 |

the dimension 16, and the latent codebook has vocabulary size 8. To model the posterior, we train a classifier, which is applied to the output of the decoder. The classifier is parameterised by a simple convolutional neural network. For on-policy training we use Log Variance divergence, and for off-policy training we additionally use a replay buffer with MCMC. The

neural network is learnt with the AdamW optimiser (Loshchilov & Hutter, 2019). Table 7 shows the hyperparameters used for training the discrete outsourced diffusion sampler.

### C.4.2. MCMC ALGORITHM

In outsourced experiments on the MNIST dataset the latent space of a generator is a discrete space $\mathcal{S} = \{1, \ldots, C\}^d$ with. $C = 8$. For this case we use a a specific MCMC which we denote *Categorical MCMC*. This MCMC algorithm can be viewed as an generalisation of *Hamming ball MCMC*, and it uses the following transition kernel, which is independent over entries $x_i$:

$$\mathcal{P}(x_i' = c' \mid x_i = c) = \begin{cases} p & c' = c, \\ \frac{1-p}{C-1} & c' \neq c. \end{cases}$$

## D. Additional Experimental Results

### D.1. Analysis

In the following subsections, we analyse some of the key algorithmic design choices of our off-policy training method, mainly under the setting of sampling in discretised synthetic distributions and Ising/Potts models.

*Table 7.* Hyperparameter settings for outsourced sampling experiments.

|  | MLP | ViT |
|---|---|---|
| Number of blocks | - | 4 |
| Number of attention heads | - | 4 |
| Hidden dim | 256 | 64 |
| Learning rate | 1e-3 | |
| Number of epochs ($N_{\text{epoch}}$) | 20000 | |
| Batch size ($M$) | 256 | |
| Buffer size | 25600 | |
| Off-to-On ratio ($R$) | 2 | |
| MCMC Sample ratio ($r$) | 0.5 | |
| MCMC Refresh Interval ($I$) | 10 | |
| MCMC steps (incl. burn-in) ($L$) | 201 | |

### D.1.1. ABLATION STUDY ON TEMPERATURE-ANNEALED TRAINING

Annealing the temperature of the target distribution throughout the training procedure has been adopted in several works in continuous diffusion samplers (Rissanen et al., 2025; Schopmans & Friederich, 2025; Akhound-Sadegh et al., 2025) and also in discrete diffusion samplers (Holderrieth et al., 2025; Zhu et al., 2025), showing powerful empirical gain over non-annealed training. We also adopt a simple temperature annealing scheme for all the learning-based algorithms, where we linearly anneal the inverse temperature from 0 to 1 over half of the training epochs.

Table 8 shows the results from each algorithm with and without the temperature annealing scheme. We observe that temperature annealing provides a consistent performance gain, as expected, while the off-policy training with MCMC exploration is less sensitive to the annealing scheme, especially in the $16 \times 16$ Ising where the MCMC algorithm mixes the modes well.

*Table 8.* Ablation study on temperature-annealed training (mean±std over 5 runs).

| Target → | 40GMM ($d = 4$) | | | | ManyWell ($d = 10$) | | | | $16 \times 16$ Ising ($\beta = 0.6$) | | | | |
|---|---|---|---|---|---|---|---|---|---|---|---|---|---|
| Method ↓ Metric → | ELBO ↑ | EUBO ↓ | MMD ↓ | Sink. ↓ | ELBO ↑ | EUBO ↓ | MMD ↓ | Sink. ↓ | ELBO ↑ | EUBO ↓ | Sink. ↓ | Mag. ↓ | Corr. ↓ |
| MDNS | | | | | | | | | | | | | |
| w/o annealing | -3.54±0.26 | 91.54±12.60 | 0.58±0.13 | 3508.04±1328.10 | 41.20±0.97 | 54.19±0.11 | 0.04±0.01 | 1.83±0.04 | 309.78±0.00 | 403.55±30.68 | 117.99±1.22 | 0.97±0.00 | 0.95±0.00 |
| w/ annealing | -16.66±2.25 | 14.02±1.50 | 0.17±0.04 | 349.31±69.18 | 41.52±0.65 | 54.16±0.14 | 0.04±0.01 | 1.82±0.05 | 310.18±0.33 | 341.82±38.37 | 48.71±55.25 | 0.41±0.46 | 0.38±0.46 |
| LV (on-policy) | | | | | | | | | | | | | |
| w/o annealing | -3.71±0.00 | 122.12±6.96 | 0.59±0.06 | 3593.82±551.92 | 48.79±0.02 | 67.20±0.83 | 0.33±0.01 | 4.81±0.20 | 309.78±0.02 | 388.39±16.48 | 116.25±2.23 | 0.97±0.00 | 0.95±0.00 |
| w/ annealing | -2.56±0.27 | 77.96±13.03 | 0.42±0.09 | 2265.84±685.58 | 49.10±0.04 | 57.40±0.26 | 0.14±0.01 | 1.34±0.02 | 309.77±0.04 | 422.53±94.60 | 116.96±2.15 | 0.97±0.00 | 0.95±0.00 |
| LV + Buffer | | | | | | | | | | | | | |
| w/o annealing | -3.51±0.39 | 91.12±7.66 | 0.56±0.06 | 3322.93±744.78 | 45.17±0.21 | 54.17±0.06 | 0.05±0.01 | 1.79±0.04 | 309.79±0.00 | 373.49±8.86 | 117.54±1.83 | 0.97±0.00 | 0.95±0.00 |
| w/ annealing | -2.68±0.19 | 49.77±11.95 | 0.38±0.10 | 1566.19±596.76 | 44.83±0.41 | 54.29±0.08 | 0.05±0.01 | 1.83±0.07 | 308.55±1.36 | 311.05±0.36 | 4.33±0.53 | 0.40±0.30 | 0.27±0.25 |
| LV + Buffer + MCMC | | | | | | | | | | | | | |
| w/o annealing | -4.43±2.99 | 45.64±18.31 | 0.48±0.15 | 2215.06±1149.08 | 43.79±0.50 | 54.19±0.11 | 0.06±0.01 | 1.97±0.06 | 308.15±1.17 | 311.21±0.32 | 4.35±0.39 | 0.37±0.24 | 0.21±0.17 |
| w/ annealing | -13.41±6.37 | 13.89±1.30 | 0.57±0.05 | 1241.69±490.85 | 42.50±0.31 | 54.44±0.02 | 0.07±0.01 | 2.16±0.03 | 310.18±0.24 | 310.65±0.08 | 3.85±0.20 | 0.14±0.13 | 0.04±0.05 |
| TB (on-policy) | | | | | | | | | | | | | |
| w/o annealing | -3.71±0.00 | 115.00±5.64 | 0.55±0.03 | 3219.80±587.75 | 48.75±0.04 | 68.14±2.13 | 0.32±0.02 | 4.84±0.30 | 307.64±0.87 | 2607.94±855.53 | 127.07±1.45 | 1.00±0.01 | 0.95±0.00 |
| w/ annealing | -2.47±0.30 | 71.53±7.55 | 0.40±0.07 | 2142.65±637.28 | 49.05±0.05 | 57.48±0.28 | 0.15±0.02 | 1.37±0.02 | 309.79±0.00 | 377.92±14.19 | 117.22±0.68 | 0.97±0.00 | 0.95±0.00 |
| TB + Buffer | | | | | | | | | | | | | |
| w/o annealing | -3.40±0.39 | 91.66±13.17 | 0.56±0.04 | 3391.08±364.60 | 48.66±0.19 | 53.31±0.14 | 0.04±0.01 | 1.33±0.02 | 309.78±0.00 | 371.93±12.93 | 117.26±1.67 | 0.97±0.00 | 0.95±0.00 |
| w/ annealing | -5.97±1.06 | 4.20±1.39 | 0.07±0.03 | 114.11±53.48 | 48.79±0.07 | 53.18±0.03 | 0.03±0.01 | 1.30±0.01 | 310.42±0.03 | 310.56±0.01 | 3.59±0.04 | 0.04±0.02 | 0.00±0.00 |
| TB + Buffer + MCMC | | | | | | | | | | | | | |
| w/o annealing | -3.05±0.52 | 52.65±12.13 | 0.41±0.12 | 1993.90±732.09 | 48.78±0.14 | 53.04±0.04 | 0.03±0.01 | 1.30±0.02 | 310.36±0.07 | 310.60±0.03 | 3.51±0.05 | 0.06±0.03 | 0.00±0.00 |
| w/ annealing | -7.04±0.88 | 1.20±0.21 | 0.05±0.01 | 9.24±4.33 | 48.73±0.18 | 52.86±0.03 | 0.04±0.01 | 1.36±0.02 | 310.43±0.01 | 310.55±0.01 | 3.47±0.12 | 0.02±0.01 | 0.00±0.00 |

### D.1.2. EFFECT OF OFF-POLICY TO ON-POLICY RATIO $R$

Here, we investigate the impact of the off-policy to on-policy training ratio $R$. While Algorithm 1 illustrates the procedure for $R \geq 1$, we can easily modify it to handle the case of $R < 1$ such that the ratio between off-policy and on-policy training is $R$.

Fig. 6 demonstrates the effect of $R$ on the performance of off-policy training for 40GMM ($d = 4 \times 8 = 32$), ManyWell ($d = 10 \times 8 = 80$), and the $16 \times 16$ Ising model with $\beta = 0.6$. In general, increasing $R$ improves EUBO but degrades ELBO. We find that values of $R$ between 1 and 4 offer a reasonable balance.

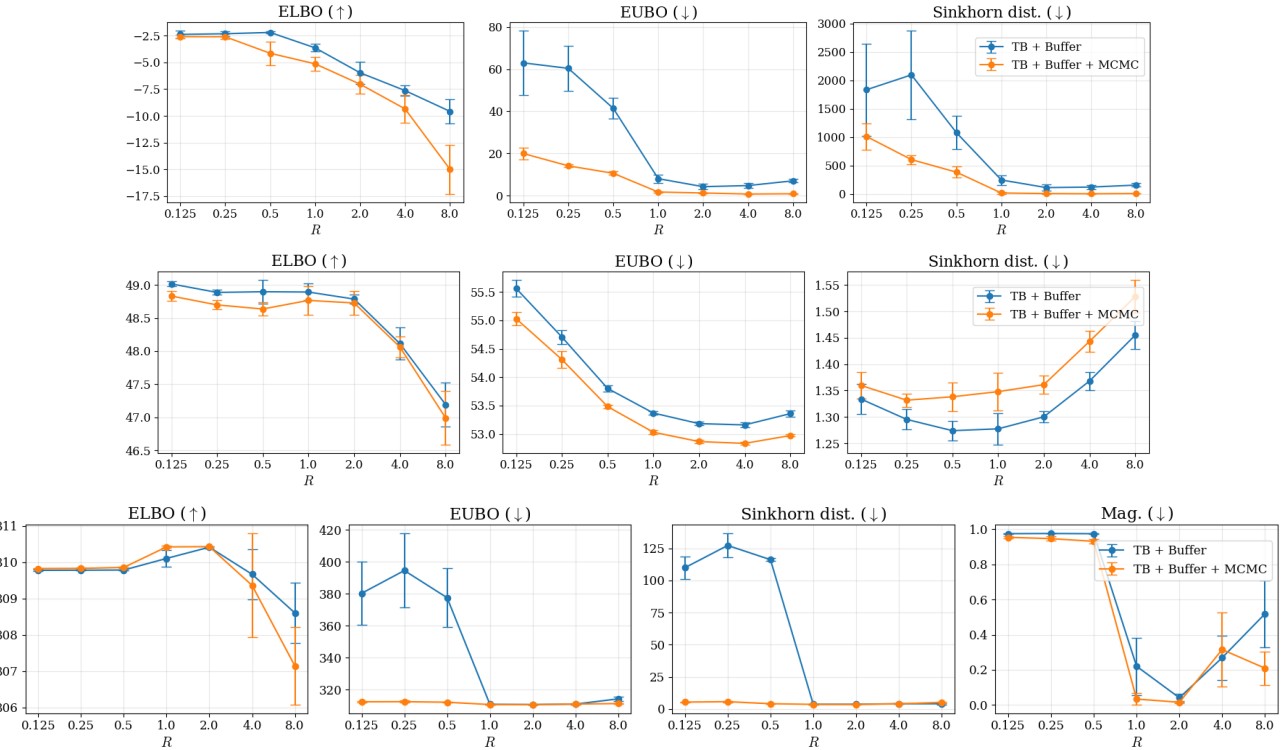

*Figure 6.* Effect of off-policy to on-policy ratio $R$ in 40GMM ($d = 4 \times 8 = 32$, top), ManyWell ($d = 10 \times 8 = 80$, mid), and $16 \times 16$ Ising ($\beta = 0.6$, bottom). The error bars show a 1-std range from 5 runs.

### D.1.3. EFFECT OF MCMC ALGORITHMS FOR ISING AND POTTS

We use Swendsen-Wang (Swendsen & Wang, 1986; 1987) MCMC, a specialised MCMC algorithm for Ising and Potts models, for off-policy exploration. Here, we replace the Swendsen-Wang with Metropolis-Hastings (MH) with 1-Hamming-ball proposal. Table 9 shows the results.

*Table 9.* Effect of MCMC Algorithms on Ising and Potts models (mean±std over 5 runs).

| Target → | $16 \times 16$ Ising ($\beta = 0.4407$) | | | $16 \times 16$ Ising ($\beta = 0.6$) | | | $16 \times 16$ Ising ($\beta = 1.2$) | | | $16 \times 16$ Potts ($\beta = 1.005$) | | | $16 \times 16$ Potts ($\beta = 1.2$) | | |
|---|---|---|---|---|---|---|---|---|---|---|---|---|---|---|---|
| Method ↓ Metric → | Sink. ↓ | Mag. ↓ | Corr. ↓ | Sink. ↓ | Mag. ↓ | Corr. ↓ | Sink. ↓ | Mag. ↓ | Corr. ↓ | Sink. ↓ | Mag. ↓ | Corr. ↓ | Sink. ↓ | Mag. ↓ | Corr. ↓ |
| MDNS | 61.60±18.23 | 0.32±0.35 | 0.23±0.26 | 48.71±55.25 | 0.41±0.46 | 0.38±0.46 | 126.27±0.90 | 1.00±0.00 | 1.00±0.00 | 84.78±10.15 | 0.08±0.06 | 0.01±0.00 | 99.95±70.52 | 0.58±0.46 | 0.01±0.00 |
| LV + Buffer | | | | | | | | | | | | | | | |
| + MH MCMC | 59.47±1.81 | 0.06±0.06 | 0.03±0.02 | 8.24±4.28 | 0.45±0.27 | 0.27±0.34 | 71.08±57.89 | 1.00±0.00 | 0.99±0.01 | 81.74±5.32 | 0.12±0.16 | 0.03±0.02 | 66.73±37.51 | 0.95±0.00 | 0.00±0.00 |
| + Swendsen-Wang | 47.34±0.90 | 0.12±0.07 | 0.06±0.01 | 3.85±0.20 | 0.14±0.13 | 0.04±0.05 | 0.04±0.04 | 0.05±0.08 | 0.01±0.02 | 84.58±1.02 | 0.04±0.03 | 0.09±0.02 | 19.56±5.21 | 0.13±0.10 | 0.02±0.02 |
| TB + Buffer | | | | | | | | | | | | | | | |
| + MH MCMC | 59.45±1.58 | 0.05±0.04 | 0.03±0.02 | 6.38±0.37 | 0.14±0.19 | 0.07±0.11 | 51.24±61.74 | 1.00±0.00 | 0.99±0.01 | 78.40±0.48 | 0.04±0.02 | 0.02±0.01 | 84.24±55.93 | 0.68±0.32 | 0.00±0.00 |
| + Swendsen-Wang | 46.22±0.36 | 0.04±0.02 | 0.02±0.01 | 3.47±0.12 | 0.02±0.01 | 0.00±0.00 | 0.02±0.00 | 0.03±0.01 | 0.00±0.00 | 78.50±0.83 | 0.03±0.02 | 0.01±0.01 | 12.37±0.82 | 0.03±0.02 | 0.00±0.00 |

### D.1.4. EFFECT OF DIFFERENT MASKING SCHEDULES

We show the result of changing the number of unmaskings per step during inference for a given trained model for the 32-dimensional 40GMM in Table 10. As expected, increasing unmaskings per step degrades performance. However, the

modest decrease in Sinkhorn distance for TB + Buffer + MCMC shows it still retains a large part of the distribuiton coverage learnt in training. The decrease in Sinkhorn distance for TB is likely due to the fact that removing inter-variable dependences by performing many unmaskings at a time overcomes some of the mode collapse in the base model, which was trained with single unmasking.

*Table 10.* Comparison of different unmasking schedules at inference time for a given trained model on 32-dimensional 40GMM.

| Method ↓ Metric → | Unmaskings per step | ELBO ↑ | EUBO ↓ | Sinkhorn ↓ |
|---|---|---|---|---|
| MDNS | 1 | $-16.66_{\pm 2.25}$ | $14.02_{\pm 1.50}$ | $349.31_{\pm 69.18}$ |
| TB | 1 | $-2.47_{\pm 0.30}$ | $71.53_{\pm 7.55}$ | $2142.65_{\pm 637.28}$ |
| | 2 | $-5.38_{\pm 1.43}$ | $71.36_{\pm 8.58}$ | $1525.84_{\pm 767.21}$ |
| | 4 | $-11.80_{\pm 5.30}$ | $70.98_{\pm 8.89}$ | $1276.40_{\pm 930.19}$ |
| | 8 | $-28.34_{\pm 13.64}$ | $69.76_{\pm 9.79}$ | $846.95_{\pm 464.03}$ |
| | 16 | $-70.30_{\pm 34.29}$ | $66.22_{\pm 12.70}$ | $584.64_{\pm 248.89}$ |
| TB+Buffer | 1 | $-5.97_{\pm 1.06}$ | $4.20_{\pm 1.39}$ | $114.11_{\pm 53.48}$ |
| | 2 | $-13.87_{\pm 1.93}$ | $4.33_{\pm 1.52}$ | $82.04_{\pm 38.18}$ |
| | 4 | $-33.12_{\pm 2.80}$ | $4.60_{\pm 1.46}$ | $65.73_{\pm 31.31}$ |
| | 8 | $-75.96_{\pm 4.41}$ | $5.13_{\pm 1.36}$ | $60.44_{\pm 19.68}$ |
| | 16 | $-172.87_{\pm 10.15}$ | $6.00_{\pm 1.04}$ | $60.74_{\pm 18.90}$ |
| TB+Buffer+MCMC | 1 | $-7.13_{\pm 0.73}$ | $0.91_{\pm 0.03}$ | $4.25_{\pm 0.30}$ |
| | 2 | $-15.09_{\pm 1.06}$ | $1.23_{\pm 0.18}$ | $14.18_{\pm 10.13}$ |
| | 4 | $-35.53_{\pm 1.70}$ | $1.62_{\pm 0.18}$ | $12.36_{\pm 1.26}$ |
| | 8 | $-78.46_{\pm 2.37}$ | $2.38_{\pm 0.17}$ | $20.19_{\pm 2.83}$ |
| | 16 | $-174.25_{\pm 5.88}$ | $3.76_{\pm 0.16}$ | $30.11_{\pm 1.94}$ |

### D.1.5. GRADIENT COMPUTATION ON PARTIAL TRAJECTORIES

The second-moment objectives we consider have the form of an L2 regression on the logits, it is therefore possible to propagate gradients only to a subset of transition log-densities on the trajectory, which would not require storing the computation graphs of the logits that do not receive gradient. This method was originally suggested in Venkatraman et al. (2024) and does not affect the expectation of the gradient estimate, while increasing its variance.

Below we show (on the 32-dimensional 40GMM) that our algorithms' performance degrades gracefully as the fraction of logits dropped out from gradient computation increases under a fixed number of training steps. All runs used the same number of training steps and we observe that the ones with a larger fraction of steps detached have not converged, which suggests that the degradation is simply due to slower training.

*Table 11.* Comparison of different fractions of logits dropped from gradient computation.

| Method ↓ Metric → | Percentage detached | ELBO ↑ | EUBO ↓ | Sinkhorn ↓ |
|---|---|---|---|---|
| MDNS | 0.0 | $-16.66_{\pm 2.25}$ | $14.02_{\pm 1.50}$ | $349.31_{\pm 69.18}$ |
| TB | 0.0 | $-2.47_{\pm 0.30}$ | $71.53_{\pm 7.55}$ | $2142.65_{\pm 637.28}$ |
| | 0.25 | $-2.62_{\pm 0.27}$ | $75.40_{\pm 14.44}$ | $2221.30_{\pm 753.55}$ |
| | 0.5 | $-2.83_{\pm 0.22}$ | $74.50_{\pm 8.58}$ | $2338.68_{\pm 703.96}$ |
| | 0.75 | $-3.01_{\pm 0.20}$ | $74.85_{\pm 5.12}$ | $2213.83_{\pm 543.56}$ |
| | 0.875 | $-11.38_{\pm 13.24}$ | $488.99_{\pm 785.44}$ | $4058.40_{\pm 1236.20}$ |
| TB+Buffer | 0.0 | $-5.97_{\pm 1.06}$ | $4.20_{\pm 1.39}$ | $114.11_{\pm 53.48}$ |
| | 0.25 | $-6.95_{\pm 0.61}$ | $5.33_{\pm 0.69}$ | $112.50_{\pm 23.28}$ |
| | 0.5 | $-7.95_{\pm 1.45}$ | $6.25_{\pm 1.92}$ | $150.03_{\pm 86.98}$ |
| | 0.75 | $-9.12_{\pm 0.78}$ | $10.23_{\pm 0.66}$ | $236.15_{\pm 38.83}$ |
| | 0.875 | $-12.57_{\pm 2.32}$ | $12.58_{\pm 2.53}$ | $293.86_{\pm 64.51}$ |
| TB+Buffer+MCMC | 0.0 | $-7.13_{\pm 0.73}$ | $0.91_{\pm 0.03}$ | $4.25_{\pm 0.30}$ |
| | 0.25 | $-8.68_{\pm 1.27}$ | $1.08_{\pm 0.17}$ | $7.44_{\pm 5.58}$ |
| | 0.5 | $-9.59_{\pm 0.86}$ | $1.79_{\pm 0.39}$ | $34.08_{\pm 18.66}$ |
| | 0.75 | $-14.37_{\pm 0.90}$ | $2.90_{\pm 0.75}$ | $54.32_{\pm 33.84}$ |
| | 0.875 | $-21.15_{\pm 3.42}$ | $3.74_{\pm 1.06}$ | $66.31_{\pm 34.73}$ |

### D.1.6. WALL-CLOCK TIME AND NUMBER OF ENERGY FUNCTION EVALUATIONS

Our experimental setup is designed to allow for a similar number of energy function evaluations (NFEs) for MDNS and off-policy algorithms (see §C). We report in Table 12 the NFEs for each algorithm below (this applies to all sampling results in Tables 1 and 5.

Since exact wall-clock times vary by hardware and target distribution, we provide a rough comparison. Algorithms with TB/LV take 1.2-1.8× longer per gradient step than MDNS. However, we trained MDNS for 2.5× more training steps, and

*Table 12.* Number of energy function evaluations (NFE) for all main algorithms compared.

| Algorithm | NFE ($\times 10^6$) |
|---|---|
| MH-MCMC | 3.533 |
| MDNS | 1.280 |
| TB/LV | 2.560 |
| TB/LV + Buffer | 0.853 |
| TB/LV + Buffer + MCMC | 1.365 |

thus its total training time is roughly 1.4-2× longer. For example, in the Potts epxeriment, TB/LV required approximately 3 hours to train, compared to approximately 4 hours for MDNS.

### D.1.7. COMPARISON WITH LEAPS

We compare our off-policy discrete diffusion sampling methods with LEAPS (Holderrieth et al., 2025). We note here that LEAPS takes significantly longer to train than both MDNS and our samplers – LEAPS takes approximately 48 hours for the Potts models, whereas MDNS and our samplers take approximately 4 hours and 3 hours, respectively (experiments were performed using a single H100).

We show the results in Table 13. On the Ising benchmark, LEAPS performs similarly to our samplers. However, on the Potts benchmark, LEAPS is worse than our off-policy samplers. Note that LEAPS results are averaged over 3 runs, while the others are over 5.

*Table 13.* Comparison of LEAPS on selected Ising and Potts models (mean±std over 3 runs for LEAPS and 5 runs for others).

| Target → | $16 \times 16$ Ising ($\beta = 0.6$) | | | $16 \times 16$ Potts ($\beta = 1.2$) | | |
|---|---|---|---|---|---|---|
| Method ↓ Metric → | Sink. ↓ | Mag. ↓ | Corr. ↓ | Sink. ↓ | Mag. ↓ | Corr. ↓ |
| LEAPS | $6.42_{\pm 0.12}$ | $0.01_{\pm 0.00}$ | $0.03_{\pm 0.00}$ | $107.27_{\pm 1.16}$ | $0.51_{\pm 0.02}$ | $0.09_{\pm 0.00}$ |
| MDNS | $48.71_{\pm 55.25}$ | $0.41_{\pm 0.46}$ | $0.38_{\pm 0.46}$ | $99.95_{\pm 70.52}$ | $0.58_{\pm 0.46}$ | $0.01_{\pm 0.00}$ |
| TB + Buffer | $3.59_{\pm 0.04}$ | $0.04_{\pm 0.02}$ | $0.00_{\pm 0.00}$ | $156.12_{\pm 2.14}$ | $0.95_{\pm 0.00}$ | $0.00_{\pm 0.00}$ |
| TB + Buffer + MCMC | $3.47_{\pm 0.12}$ | $0.02_{\pm 0.01}$ | $0.00_{\pm 0.00}$ | $12.37_{\pm 0.82}$ | $0.03_{\pm 0.02}$ | $0.00_{\pm 0.00}$ |

## D.2. Visualisations

Here we provide additional visualisation of samples in addition to the ones in the main text.

### D.2.1. ISING AND POTTS MODEL

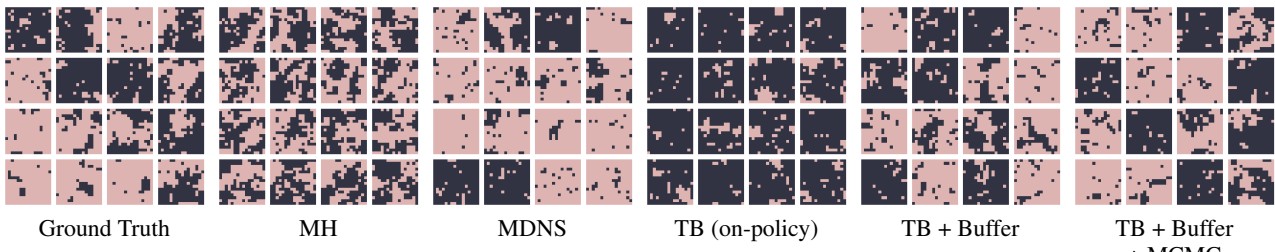

| Ground Truth | MH | MDNS | TB (on-policy) | TB + Buffer | TB + Buffer + MCMC |

*Figure 7.* Visualisation of 16 samples generated by each method for the $16 \times 16$ Ising model with $\beta = 0.4407$. The samples are obtained from the first run (seed 0).

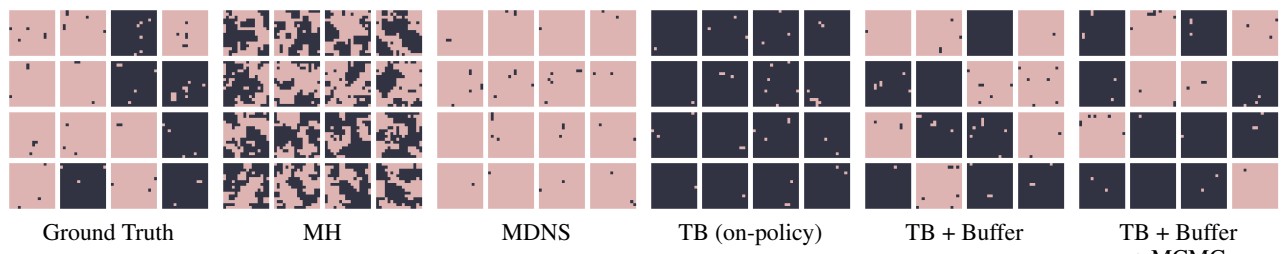

| Ground Truth | MH | MDNS | TB (on-policy) | TB + Buffer | TB + Buffer + MCMC |

*Figure 8.* Visualisation of 16 samples generated by each method for the $16 \times 16$ Ising model with $\beta = 0.6$. The samples are obtained from the first run (seed 0).

### D.2.2. DISCRETISED SYNTHETIC DENSITIES

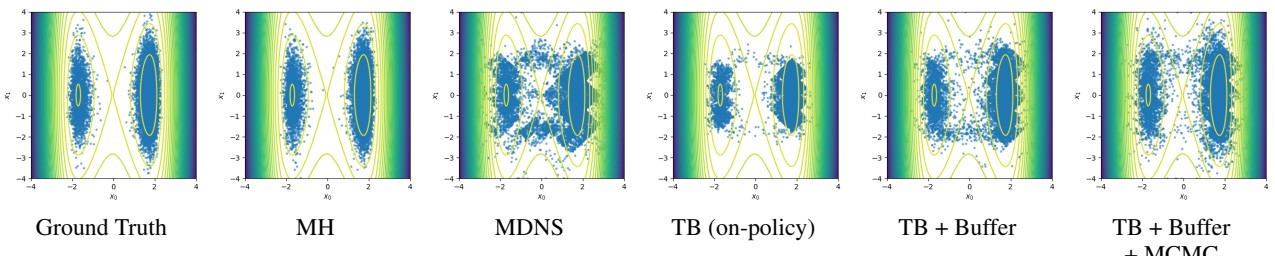

| Ground Truth | MH | MDNS | TB (on-policy) | TB + Buffer | TB + Buffer + MCMC |

*Figure 9.* Visualisation of samples generated by each method for the discretised ManyWell ($d = 10 \times 8 = 80$). The samples are obtained from the first run (seed=0) and projected to the first two dimensions.

### D.2.3. OUTSOURCED DISCRETE DIFFUSION SAMPLING EXPERIMENTS ON MNIST

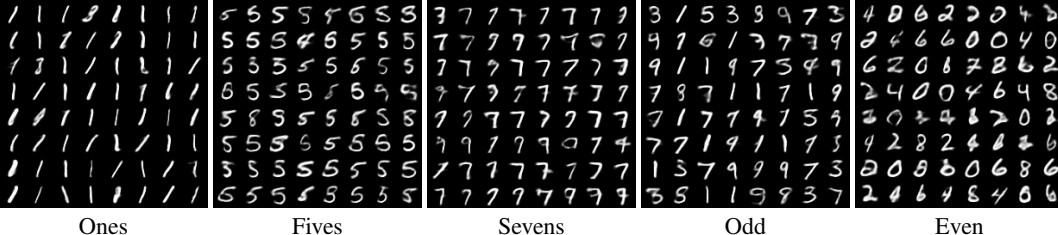

Ones      Fives      Sevens      Odd      Even

*Figure 10.* Visualisations of decoded samplers of a discrete diffusion sampler in latent space of a VQ-VAE trained on MNIST. The diffusion sampler is parameterised by MLP model, on-policy method (LV) is used for training.

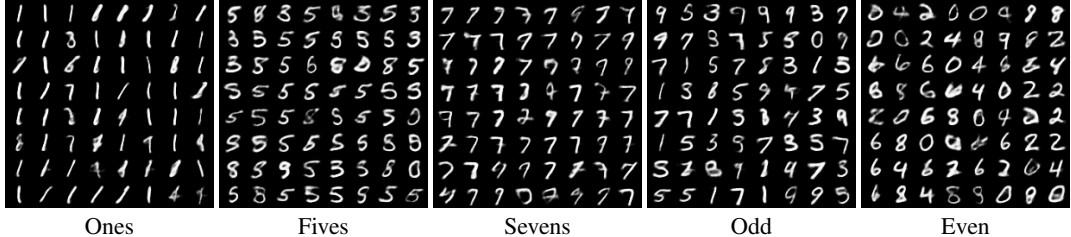

Ones      Fives      Sevens      Odd      Even

*Figure 11.* Visualisations of decoded samplers of a discrete diffusion sampler in latent space of a VQ-VAE trained on MNIST. The diffusion sampler is parameterised by MLP model, off-policy method (LV + Buffer + MCMC) is used for training.

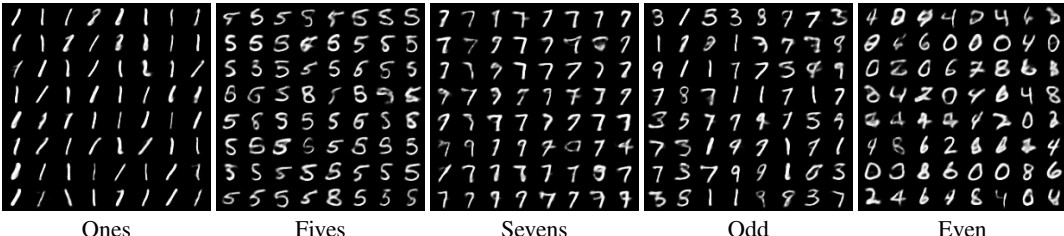

Ones      Fives      Sevens      Odd      Even

*Figure 12.* Visualisations of decoded samplers of a discrete diffusion sampler in latent space of a VQ-VAE trained on MNIST. The diffusion sampler is parameterised by ViT model, on-policy method (LV) is used for training.

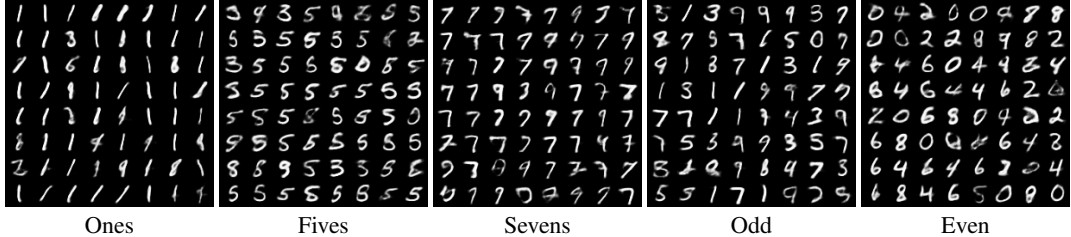

Ones      Fives      Sevens      Odd      Even

*Figure 13.* Visualisations of decoded samplers of a discrete diffusion sampler in latent space of a VQ-VAE trained on MNIST. The diffusion sampler is parameterised by ViT model, off-policy method (LV + Buffer + MCMC) is used for training.

