# OpenReview forum: "Discrete Diffusion Samplers and Bridges: Off-Policy Algorithms and Applications in Latent Spaces"
_ICML.cc/2026/Conference — ICML 2026 regular_

### Official Review · Reviewer_Kx8k · 2026-03-11

**Soundness:** 3
**Presentation:** 3
**Significance:** 2
**Originality:** 3
**Overall Recommendation:** 4
**Confidence:** 3

**Summary:**

The central problem investigated by the paper is how to train effective diffusion-based samplers for discrete unnormalized target distributions, where only the energy function is available and direct sampling is hard. The paper proposes to improve recent discrete diffusion samplers by introducing off-policy training strategies, including replay buffers, importance-weighted replay, and MCMC refinement

**Compliance With Llm Reviewing Policy:**

Affirmed.

**Key Questions For Authors:**

For the posterior sampling application, the evaluation is still somewhat limited. The paper mainly shows decoded visual samples for conditions like odd, even, or a single digit on MNIST, while it would be good to have further challenging datasets to demonstrate. Or can the author validate that using a dataset such as MNIST is good enough?

**Limitations:**

See Key Question and Weaknesses.

**Strengths And Weaknesses:**

The paper has a clear goal. Discrete diffusion sampling is still relatively underdeveloped compared with the continuous case, so it is useful to bring in training ideas that already helped continuous diffusion samplers. The off-policy methods are easy to understand at a high level, and the paper gives a coherent story for why replay and MCMC refinement can help exploration and mode coverage. The empirical results are also fairly convincing on the main synthetic tasks.

However, the paper’s strongest evidence is still on controlled synthetic benchmarks. The main results on the picked datasets are useful and standard for this topic, but they do not fully show how the method behaves on larger or more realistic discrete domains.

---

> ### Author Rebuttal · Authors · 2026-03-30
>
> Thank you for your constructive feedback.
>
> ## Scale and difficulty of benchmarks
>
> We used synthetic densities for benchmarking because they are the only targets that have been studied in past work on discrete diffusion-based sampling, notably MDNS. Our proposed algorithms perform strongly compared to baselines on these previously considered problems.
>
> We also introduced the discretised 2-dimensional synthetic densities from past work on discrete-space probabilistic inference. These densities do not have the sparse pairwise potential structure of the physics models and instead have complex dependences between variables.
>
> In addition, we introduced a more difficult setting -- the VQ-VAE latent space posterior sampling -- which is a setting that has not been studied before in the literature and involves inference under a data-derived prior. We believe that this is an important step towards showing the applicability of these methods to more realistic settings.

---

### Official Review · Reviewer_K48s · 2026-03-11

**Soundness:** 4
**Presentation:** 3
**Significance:** 3
**Originality:** 2
**Overall Recommendation:** 4
**Confidence:** 4

**Summary:**

This paper addresses the question of learning a) a sampling process for an unnormalised density $e^{-E(x)}$ over a discrete domain and b) a bridge process between samples from some distribution $p_0(x)$ and an unnormalised density $e^{-E(x)}$. The overall strategy is the minimisation of the log-variance loss which allows off-policy training of the "forward" policy which transforms samples from $p_0(x)$ into $e^{-E(x)}$. When learning the bridge process, the authors additionally learn a backwards process from $e^{-E(x)}$ to samples from $p_0(x)$ by maximising the log-likelihood of the backward kernels on the policy of the forward model. The authors claim that the paper establishes a novel connection between GFlowNets and masking diffusion models, and that they bring multiple techniques from continuous diffusion samplers to the discrete domain. The approach is tested on the sampling from the energy of a binary Potts model and Gray-code discretised two-dimensional synthetic densities. They furthermore test the approach on conditional image generation via a VQ VAE with discrete latent distribution. Finally, they test the bridge process on pairs of Gray-code discretised synthetic two-dimensional densities.

**Compliance With Llm Reviewing Policy:**

Affirmed.

**Final Justification:**

The detailed discussion of the authors addressed my mentioned weaknesses and questions, and I would recommend this paper for acceptance to the AC. However, I remain reserved about the degree of significance, novelty and empirical evaluation of the contributions. Since the paper is already heading towards acceptance, I would therefore like to maintain my score.

**Key Questions For Authors:**

1. In summary, what are the differences between the trajectory balance off-policy training and the training of GFlowNets? What are methodologically the key contributions in this paper?
2. Is it also possible to train on partial trajectories, only? What would be the main drawbacks in doing so?

**Limitations:**

Yes

**Strengths And Weaknesses:**

Strengths:
- The paper is well written and provides extensive references to the related literature.
- The formulation of discrete diffusion samplers in terms of the log variance loss is elegant and transparent, and in my opinion more accessible than formulations of GFlowNets.
- The paper is supported by a diverse set of experiments.

Weaknesses:
- I am not entirely convinced of the originality of this work. The authors say themselves that GFlowNets are an off-policy training method for sampling processes with trajectory balance loss, and off-policy TB training appears to perform the best on many benchmarks. I would appreciate if contributions made in this paper could be highlighted more clearly. I also don't share the opinion that the connection between RL and diffusion-based samplers are novel, see e.g. Sangwoong et al. Maximum Entropy Inverse Reinforcement Learning of Diffusion Models with Energy-Based Models, and Sangwoong et al. Value Gradient Sampler: Sampling as Sequential Decision Making for similar connections in the continuous domain. Noticing these connections in the discrete domain does not feel like a big step.
- The training loop requires in each iteration a whole trajectory for training. This limitation may be shared across methods in this field, but could render training prohibitively expensive for large state spaces, especially when the target distribution has sparse energy minima.
- Experiments are dimensionally small (16x16 for Ising and Potts models, 16 or 32 dimensions for synthetic densities, and a 16-dimensional 8 word codebook for the VQ VAE). For these state space sizes, brute force sampling with Gibbs with Gradients or discrete Langevin, and in the case of the VQ VAE learning an autoregressive sampling policy seems feasible. Less tractable settings would make the case for a diffusion sampler more convincing.
- Having the training algorithm in the main text would be helpful to understand the different elements of the suggested methodology.
- Personally, I find a related work section instead of interleaved with text helpful when reviewing the contributions of a paper. (No impact on my score)
- Regarding the conditional sampling I would remove the Bayesian terminology, i.e. refer to marginal and conditional distributions instead of prior and posterior.

---

> ### Author Rebuttal · Authors · 2026-03-30
>
> Thank you for your constructive feedback.
>
> ## Originality and prior work
>
> We agree that the connection to max-ent RL is not new and existed in prior works, both in the diffusion setting (such as in the work you mention) and the discrete one (for example, the papers connecting GFlowNets and RL [Deleu et al., 2024] and [Tiapkin et al., 2024] that we cite). However, the use of these connections to bring ideas from those areas to sampling with discrete diffusions is new, as is the bridge generalisation and new sampling problems we introduce in the present work. We show that introducing such ideas gives flexibility (e.g., to do multiple unmasking, uniform noising, bridge problems) that is impossible under previous frameworks.
>
> For more discussion, please see the response to Reviewer jFXW, "Prior work and novelty".
>
> ## Memory usage and partial trajectories
>
> This is an important point to consider when scaling samplers. Our algorithms, as do those in other work, need full-trajectory rollouts to update the parameters. While this was not a limiting factor in our experiments, optimisations to reduce memory usage are possible and could become necessary in larger problems.
>
> First, subtrajectory objectives are possible, although they require the learning of intermediate marginal densities (cf. [Madan et al, "Learning GFlowNets from partial episodes...", 2023]).
>
> Second, because the second-moment objectives we consider have the form of an L2 regression on the logits, it is possible to propagate gradients only to a subset of transition log-densities on the trajectory, which would not require storing the computation graphs of the logits that do not receive gradient. This method was originally suggested in [Venkatraman et al., "Amortizing intractable inference in diffusion models", 2024; §H.1] and does not affect the expectation of the gradient estimate, while increasing its variance. Below we show (on the 32-dimensional 40GMM) that our algorithms' performance degrades gracefully as the fraction of logits dropped out from gradient computation increases under a fixed number of training steps. All runs used the same number of training steps and we observe that the ones with a larger fraction of steps detached have not converged, which suggests that the degradation is simply due to slower training.
>
> |Algorithm|Percentage detached|ELBO|EUBO|Sinkhorn|
> |--|:--:|:--:|:--:|:--:|
> |MDNS|0.0|-16.66±2.25|14.02±1.50|349.31±69.18|
> |TB|0.0|-2.47±0.30|71.53±7.55|2142.65±637.28|
> |TB|0.25|-2.62±0.27|75.40±14.44|2221.30±753.55|
> |TB|0.5|-2.83±0.22|74.50±8.58|2338.68±703.96|
> |TB|0.75|-3.01±0.20|74.85±5.12|2213.83±543.56|
> |TB|0.875|-11.38±13.24|488.99±785.44|4058.40±1236.20|
> |TB+Buffer|0.0|-5.97±1.06|4.20±1.39|114.11±53.48|
> |TB+Buffer|0.25|-6.95±0.61|5.33±0.69|112.50±23.28|
> |TB+Buffer|0.5|-7.95±1.45|6.25±1.92|150.03±86.98|
> |TB+Buffer|0.75|-9.12±0.78|10.23±0.66|236.15±38.83|
> |TB+Buffer|0.875|-12.57±2.32|12.58±2.53|293.86±64.51|
> |TB+Buffer+MCMC|0.0|-7.13±0.73|0.91±0.03|4.25±0.30|
> |TB+Buffer+MCMC|0.25|-8.68±1.27|1.08±0.17|7.44±5.58|
> |TB+Buffer+MCMC|0.5|-9.59±0.86|1.79±0.39|34.08±18.66|
> |TB+Buffer+MCMC|0.75|-14.37±0.90|2.90±0.75|54.32±33.84|
> |TB+Buffer+MCMC|0.875|-21.15±3.42|3.74±1.06|66.31±34.73|
>
> ## Difficulty and scale of experiments
>
> Please see the end of the response to Reviewer Kx8k below regarding the difficulty of the benchmarks in comparison to those considered in recent work.
>
> ## Algorithm in main text, related work section
>
> Thank you for the suggestions. Given an extra page in the camera-ready version, we can easily extract the related work discussed throughout the text in to a separate section and include an algorithm box.
>
> ## Bayesian and marginal/conditional terminology
>
> The two terminologies are equivalent for our purposes, since a Bayesian posterior $p(x\mid y)$ is the conditional of a joint $p(x,y)$ and the prior $p(x)$ is its marginal. We chose the Bayesian language because it is quite standard in the setting of inverse problems and links our work with the large body of past work on posterior sampling and amortised inference under data-derived diffusion priors for Bayesian inverse problems in the continuous setting (e.g., [Kadkhodaie et al, "Solving linear inverse problems...", 2021], [Kawar et al., "SNIPS...", 2021], [Chung et al., "Diffusion posterior sampling...", 2023], the [Venkatraman et al., 2024] cited above, and many others). In that setting, marginal/conditional are less intuitive: viewing the data-derived prior as a marginal distribution over the observations, where we marginalise over possible conditions, requires us to assume some distribution over conditions $y$, which is not something the inference algorithm controls or approximates.

---

> > ### Author Rebuttal · Reviewer_K48s · 2026-04-03
> >
> > I thank the authors for their detailed discussion of the mentioned weaknesses and questions. Since the paper is already heading towards acceptance, and I remain reserved about the strength of this paper regarding novelty and empirical evaluation (I appreciate the discussion in the author response), I would like to maintain my score.

---

### Official Review · Reviewer_jFXW · 2026-03-12

**Soundness:** 3
**Presentation:** 4
**Significance:** 3
**Originality:** 3
**Overall Recommendation:** 5
**Confidence:** 3

**Summary:**

This paper proposes a framework to train train discrete generative samplers using trajectory-level consistency objectives. Indeed, the samplers are trained by matching the full probability path. This formulation led the authors to introduce off-policy training techniques for discrete state space. They also use it to generalize bridge matching between two distributions in the discrete state space case. On the experimental side, the method is tested and compared against others on a variety of instances, from toy experiments to applications in latent space.

**Compliance With Llm Reviewing Policy:**

Affirmed.

**Final Justification:**

Despite some novelty concerns, my questions are resolved, and my final assessment remains positive.

**Key Questions For Authors:**

I have three main questions.

1. The paper itself says that trajectory balance and off-policy training were already used in earlier work, citing for example GflowNets. Correct me if I am wrong, but to my knowledge GflowNets were developed in discrete state space. Can the authors precisely articulate what is methodologically new in the paper relative to prior GFlowNets or other discrete sampling work that already used trajectory balance and off-policy training. Then, can they identify which empirical gains come specifically from theses new ingredients?

2. Following that point, it is currently unclear to me which ingredients in the current manuscript are responsible for the gains, and in particular their relative costs. Can you insist on disentangling the impact of the TB loss, then the TB + buffer, then TB + buffer + MCMC, but also by adding some costs, or by explicitly mentioning whether the methods are tested under matched wall-clock/compute or not? I think that the paper would beneficiate from that clarification.

3. And following the main weakness I pointed out : do you know how is the method working when scaled against current paradigms (i.e. working with $C \approx 10^5$ and $d \approx 10^4$), e.g. dLLMs that are trained by maximizing a variational ELBO ?

Despite all these questions, I think that the paper is good overall. I am quite clearly in favor for acceptance, as the strengths of the paper are solid. I think that the paper opens several questions for future works.

**Limitations:**

Yes.

**Strengths And Weaknesses:**

- Strenghts ;

  - The paper treats an understudied problem in the area of discrete generative modeling. The approach of bringing and generalizing known techniques from continuous spaces to discrete state spaces is interesting.
  - The paper is really well written, and it's quite clear to understand the backgrounds and the problem it aims to tackle.
  - Empirical results are interesting ; in particular, the method seems to constantly outperforms MDSN, a main concurrent method previously proposed.


- Weaknesses ;

  - The main weakness I can see is that, despite numerical interesting results (from toy physical expriments to latent spaces), the method is not compared against the current paradigm of generative modeling in discrete state spaces ( e.g. diffusion language models). Indeed, it is unclear to see how well the methods perform for generative modeling tasks in really large discrete state spaces (with e.g. $C \approx 10^5$ and $d \approx 10^4$).

---

> ### Author Rebuttal · Authors · 2026-03-30
>
> Thank you for your constructive feedback.
>
> ## Comparison to large-scale generative modelling
>
> To avoid misunderstanding, the problem considered in discrete diffusion language modelling is quite different from the one we consider. In discrete diffusion language modelling, the goal is to learn a generative model of a dataset by maximising (a variational bound on) data log-likelihood. In our setting, the goal is to learn an amortised sampler for a target distribution that we can query through an unnormalised density. The two problems -- generative modelling vs. variational inference or sampling -- require different algorithms and are not directly comparable.
>
> In comparison to generative modelling given a dataset, the sampling setting:
> - is generally much more difficult (as no information about the modes of the distribution to be approximated is given);
> - requires different families of algorithms (reinforcement learning and control-based algorithms, including GFlowNets, are used for amortised sampling, while likelihood maximisation or proxies are used for generative modelling);
> - has different applications: while generative models can produce samples resembling those from the data distribution, posterior sampling under data-derived priors can be used for Bayesian inverse problems, sampling from energy-based models, etc.
>
> Please also see the response to Reviewer Kx8k about the difficulty of the problems considered in the context of discrete-space sampling and the next response on the novelty of the paper in the context of other sampling work.
>
> ## Prior work and novelty
>
> In the current state of the field, sampling (using RL and control-based algorithms), diffusion modelling, and sequential amortised inference (using GFlowNets, for instance) are often seen as three separate families of algorithms, and the problems they address are often seen as separate. We believe that it is very important and beneficial for the community to show the connections between these approaches and to transfer useful findings between them.
>
> While indeed part of the novelty is conceptual -- executing this transfer and showing the connections -- we also brought together some ideas that were not brought together before:
> - From GFlowNets to sampling:
>   - the use of off-policy methods, including the use of MCMC-updated buffers, for diffusion-based sampling in discrete spaces;
>   - training objectives that are more general than approximate forward KL and log-variance objectives, which are most elegantly stated in the GFlowNet language.
> - From discrete diffusion modelling to sampling:
>   - the use of multiple unmasking, while previous work, written both in the GFlowNet language (e.g., the mentioned [Zhang et al., 2022a]) and in the diffusion language considered single-token unmasking;
>   - the use of uniform diffusion (while previous sampling work considered unmasking), which relies on the bridge formulation.
>
> In addition to this conceptual novelty, the unique novelties of this paper include:
> - The introduction of latent-space inference (noise optimisation / outsourced sampling) for posterior inference under discrete latent variable model priors, which is a setting that has not been considered before in the literature;
> - The introduction of discrete data-to-energy Schrödinger bridges as an amortised sampling problem, also never considered before (of note, learning of bridges is not naturally expressed in the GFlowNet language);
> - Evaluation of discrete diffusion samplers on new and more difficult problems than in past work, including harder synthetic benchmarks and the outsourced sampling application.
>
> **Regarding your question on GFlowNet sampling algorithms:** Although GFlowNets were indeed introduced in discrete spaces in 2021, they were generalised to continuous spaces over three years ago [Lahlou et al., 2023], and this generalisation has been used in a large number of subsequent works, including both diffusion-based sampling  (such as [Sendera et al., 2024],  [Choi et al., 2025], [Gritsaev et al., 2025] cited in our submission)  and other settings, including hybrid continuous/discrete sampling ([Hernández-García et al., "Crystal-GFN...", 2023], [Deleu et al., "Joint Bayesian inference...", 2023], [Zhou et al., "PhyloGFN...", 2024], [Phillips et al., "MetaGFN...", 2024], [da Silva et al., "Streaming Bayes GFlowNets", 2024], [Boussif et al., "Bayesian symbolic regression...", 2025]).
>
> ## Costs and effects of algorithm components
>
> Regarding which ingredients are responsible for the gains: If the question is about the effect of the main new ingredients (buffer and MCMC), then we *already* report TB, TB+buffer, TB+buffer+MCMC, in exactly that order, in Tables 1 and 2.
>
> Methods are matched not in wall clock but in energy evaluations, which is the real bottleneck in amortisation. This cost is incurred at training, not at inference. Please see the response to Reviewer S5kg ("Wall-clock time and number of energy evaluations").

---

> > ### Author Rebuttal · Reviewer_jFXW · 2026-04-02
> >
> > Thank you for the detailed rebuttal. My main questions are now resolved.
> >
> > In particular, the clarification on the intended problem setting helped: I agree that the paper addresses amortized sampling / inference for unnormalized target distributions, which is different from likelihood-based large-scale discrete generative modeling, so my earlier comparison to diffusion language models was not the most appropriate one.
> >
> > I also acknowledge that the paper already reports the progression TB, TB + buffer, and TB + buffer + MCMC in the experimental tables. My question, however, was specifically about wall-clock cost rather than energy evaluations. That said, I acknowledge the additional table provided in the response to Reviewer S5kg, and I encourage the authors to include this comparison in the revised version.
> >
> > That said, I still retain a mild reservation regarding novelty. The rebuttal clarified several meaningful contributions, but my impression remains that part of the paper’s contribution is conceptual, in the sense of connecting and transferring ideas that were already present in prior discrete-space sampling / GFlowNet-style work. I do think the paper contains worthwhile new ingredients and is overall a good paper, but I still see the novelty as somewhat less clear-cut than the empirical and presentation strengths.
> >
> > Overall, my questions are resolved, and my final assessment remains positive.

---

### Official Review · Reviewer_S5kg · 2026-03-12

**Soundness:** 3
**Presentation:** 4
**Significance:** 3
**Originality:** 3
**Overall Recommendation:** 5
**Confidence:** 4

**Summary:**

This paper studies sampling from discrete unnormalized distributions when target samples are unavailable. Its main technical contribution is to bring off-policy RL ideas into discrete diffusion sampling via replay buffers, importance-weighted buffer sampling, and MCMC refinement of buffered terminal states. The diffusion sampler is trained with second-moment objectives like trajectory balance (TB) and log-variance (LV). The paper also evaluates the framework in discrete data-to-energy settings, where one side is given by samples and the other by an unnormalized density, and it explores posterior sampling in the discrete latent spaces of pretrained VQ-VAE models.

**Compliance With Llm Reviewing Policy:**

Affirmed.

**Final Justification:**

This paper studies sampling from discrete unnormalized distributions when target samples are unavailable. Its main technical contribution is to bring off-policy RL ideas into discrete diffusion sampling via replay buffers, importance-weighted buffer sampling, and MCMC refinement of buffered terminal states. The empirical study on the main sampling tasks is strong and the experiments go beyond the usual Ising/Potts setup and beat previous approaches. I recommend acceptance.

**Key Questions For Authors:**

1. Could the authors clarify whether a comparison is infeasible or not like-for-like with LEAPS?
2. How much of the improvement at low temperature comes from off-policy learning itself vs Swendsen-Wang for Ising/Potts?
3. How sensitive are results to the masking schedule and to the number of sampling steps used at test time?

**Limitations:**

yes

**Strengths And Weaknesses:**

# Strengths

* The paper is very well written and easy to follow. I especially liked Figure 1: it communicates the path-space view clearly and is also visually memorable. More broadly, the paper does a nice job connecting discrete diffusion, stochastic control, and prior off-policy / GFlowNet.

* The empirical study on the main sampling tasks is strong and the experiments go beyond the usual Ising/Potts setup. That latent-space application felt novel and interesting to me. Same for the  data-to-energy bridge that broadens the paper beyond plain sampling.

# Weaknesses

* Part of the novelty is conceptual rather than algorithmic. The paper itself notes that in the single-unmasking case the framework recovers Zhang et al. (2022a).

* Baseline coverage could be stronger. The main learned baseline is MDNS. LEAPS is excluded entirely. As a result, the paper does not fully establish where it stands relative to the strongest current discrete neural samplers.

* The paper notes rollout/storage cost as a limitation, but does not quantify wall-clock or energy-evaluation time costs.

* The paper doesn't report ESS or other weight degeneracy diagnostic.

---

> ### Author Rebuttal · Authors · 2026-03-30
>
> Thank you for your constructive feedback.
>
> ## Novelty is conceptual rather than algorithmic
>
> Please see the response to Reviewer jFXW, "Prior work and novelty".
>
> ## Baseline coverage
>
> The results from MDNS [Zhu et al., 2025] on the Ising and Potts models show that MDNS outperforms LEAPS on these problems and is thus a more competitive method for comparison, so we did not initially run LEAPS.
>
> **We have now run LEAPS on the Ising and Potts benchmarks using the default configs provided in their repo for 100K training steps. We found that it is both an order of magnitude slower than other methods and does not perform as well.**
> - LEAPS training takes ~48 hours on Potts models, whereas MDNS and our samplers take ~6 hours and ~3 hours, respectively (we used a single H100).
> - On the Ising benchmark, the methods perform similarly to our samples. However, on the Potts, LEAPS is worse than our off-policy samplers. See the tables below.
>
> LEAPS results are averaged over 3 runs, while others are over 5.
>
> **$16 \times 16$ Ising ($\beta=0.6$)**
>
> |Algorithm|Sink.↓|Mag.↓|Corr.↓|
> |--|:--:|:--:|:--:|
> |LEAPS|6.42±0.12|0.01±0.00|0.03±0.00|
> |MDNS|48.71±55.25|0.41±0.46|0.38±0.46|
> |TB+Buffer|3.59±0.04|0.04±0.02|0.00±0.00|
> |TB+Buffer+MCMC|3.47±0.12|0.02±0.01|0.00±0.00|
>
> **$16 \times 16$ Potts ($\beta=1.2$)**
>
> |Algorithm|Sink.↓|Mag.↓|Corr.↓|
> |--|:--:|:--:|:--:|
> |LEAPS|107.27±1.16|0.51±0.02|0.09±0.00|
> |MDNS|99.95±70.52|0.58±0.46|0.01±0.00|
> |TB+Buffer|156.12±2.14|0.95±0.00|0.00±0.00|
> |TB+Buffer+MCMC|12.37±0.82|0.03±0.02|0.00±0.00|
>
> ## Wall-clock time and number of energy evaluations
>
> Our experimental setup is designed to allow for a similar number of function (energy) evaluations (NFE) for MDNS and off-policy algorithms (see C.1.2 \& C.1.4 for details). We report the NFEs for each algorithm below (this applies to all sampling results in Tables 1-2):
>
> |Algorithm|NFE ($\times10^6$)|
> |--|:--:|
> |MH-MCMC|3.533|
> |MDNS|1.280|
> |TB/LV|2.560|
> |TB/LV+Buffer|0.853|
> |TB/LV+Buffer+MCMC|1.365|
>
> Since exact wall-clock times vary by hardware and target distribution, we provide a rough comparison. Algorithms with TB/LV take 1.2-1.8x longer per gradient step than MDNS. However, we trained MDNS for 2.5x more training steps, and thus its total training time is roughly 1.4-2x longer. For example, in the Potts experiment (with Nvidia L40s), TB/LV required ~3 hours, whereas MDNS required ~4 hours.
>
> ## ESS and weight degeneracy
>
> We decided to omit ESS from the evaluation metric since it is not a good metric for mode coverage (see, e.g., [Elvira et al., "Rethinking ESS...", 2022] and [Blessing et al., "Beyond ELBOs...", 2024]). The table below provides ESS results for the 32-dimensional GMM40.
>
> |Algorithm|ESS|EUBO|
> |--|:--:|:--:|
> |MDNS|0.18±0.07|14.02±1.50|
> |TB|0.73±0.15|71.53±7.55|
> |TB+Buffer|0.15±0.06|4.20±1.39|
> |TB+Buffer+MCMC|0.12±0.07|0.91±0.03|
>
> Given that on-policy TB shows severe mode-collapse (see EUBO and Fig. 3 as well), higher ESS does not indicate better sampling quality, especially when the target distribution is highly multi-modal.
>
> ## Improvement from MCMC vs. off-policy learning
>
> This is a good question. In fact, **we have already presented results on this in Appendix D.1.3**, where we show that the use of a simpler MCMC gives some improvement, even though it is not as large with the faster-converging Swendsen-Wang algorithm. We already point to this appendix in §4.1, but will make it more prominent in the main text.
>
> ## Sensitivity to masking schedule
>
> In the table below (experiments on 32-dimensional 40GMM), we show the results from changing the number of unmaskings per step during inference for a given trained model. As expected, increasing unmaskings per step degrades performance. However, the modest decrease in Sinkhorn distance for TB + Buffer + MCMC shows it still retains a large part of the distribution coverage learnt in training. The decrease in Sinkhorn distance for TB is likely due to the fact that removing inter-variable dependences by performing many unmaskings at a time overcomes some of the mode collapse in the base model, which was trained with single unmasking.
>
> |Algorithm|Unmaskings per step|ELBO|EUBO|Sinkhorn|
> |--|:--:|:--:|:--:|:--:|
> |MDNS|-|-16.66±2.25|14.02±1.50|349.31±69.18|
> |TB|1|-2.47±0.30|71.53±7.55|2142.65±637.28|
> |TB|2|-5.38±1.43|71.36±8.58|1525.84±767.21|
> |TB|4|-11.80±5.30|70.98±8.89|1276.40±930.19|
> |TB|8|-28.34±13.64|69.76±9.79|846.95±464.03|
> |TB|16|-70.30±34.29|66.22±12.70|584.64±248.89|
> |TB+Buffer|1|-5.97±1.06|4.20±1.39|114.11±53.48|
> |TB+Buffer|2|-13.87±1.93|4.33±1.52|82.04±38.18|
> |TB+Buffer|4|-33.12±2.80|4.60±1.46|65.73±31.31|
> |TB+Buffer|8|-75.96±4.41|5.13±1.36|60.44±19.68|
> |TB+Buffer|16|-172.87±10.15|6.00±1.04|60.74±18.90|
> |TB+Buffer+MCMC|1|-7.13±0.73|0.91±0.03|4.25±0.30|
> |TB+Buffer+MCMC|2|-15.09±1.06|1.23±0.18|14.18±10.13|
> |TB+Buffer+MCMC|4|-35.53±1.70|1.62±0.18|12.36±1.26|
> |TB+Buffer+MCMC|8|-78.46±2.37|2.38±0.17|20.19±2.83|
> |TB+Buffer+MCMC|16|-174.25±5.88|3.76±0.16|30.11±1.94|

---

> > ### Author Rebuttal · Reviewer_S5kg · 2026-04-04
> >
> > The authors addressed my concerns. I think that the paper is complete with the inclusion of LEAPS. I recommend acceptance

---

### Decision · Program_Chairs · 2026-04-30

**Decision:**

Accept (regular)

**Comment:**

This paper proposes a new discrete diffusion sampler by using off-policy RL techniques. All reviewers acknowledge the clear motivation and clean presentation, and the proposed discrete neural sampler is technically sound. The experiments cover a range of tasks, including discrete latent spaces of pretrained VQ-VAEs.
During the rebuttal, the authors addressed most concerns, especially adding the LEAPS baseline and incorporating additional metrics such as ESS and NFE.

The remaining concerns center on novelty and significance. Multiple reviewers noted similarities to existing work, particularly GFlowNets for discrete distributions and RL-based approaches for continuous diffusion samplers. The significance of the proposed extension is not clear. Besides, the empirical analysis is largely limited to small-scale, low-dimensional models.